# Guided Diffusion Sampling on Function Spaces with Applications to PDEs

**Jiachen Yao**,[*]  **Abbas Mammadov**[*][†]
California Institute of Technology

**Julius Berner**
NVIDIA

**Gavin Kerrigan**
University of Oxford

**Jong Chul Ye**
KAIST

**Kamyar Azizzadenesheli**
NVIDIA

**Anima Anandkumar**
California Institute of Technology

## Abstract

We propose a general framework for conditional sampling in PDE-based inverse problems, targeting the recovery of whole solutions from extremely sparse or noisy measurements. This is accomplished by a function-space diffusion model and plug-and-play guidance for conditioning. Our method first trains an unconditional discretization-agnostic denoising model using neural operator architectures. At inference, we refine the samples to satisfy sparse observation data via a gradient-based guidance mechanism. Through rigorous mathematical analysis, we extend Tweedie's formula to infinite-dimensional Banach spaces, providing the theoretical foundation for our posterior sampling approach. Our method (FunDPS) accurately captures posterior distribution in function spaces under minimal supervision and severe data scarcity. Across five PDE tasks with only 3% observation, our method achieves an average 32% accuracy improvement over state-of-the-art fixed-resolution diffusion baselines while reducing sampling steps by 4x. Furthermore, multi-resolution fine-tuning ensures strong cross-resolution generalizability and speedup. To the best of our knowledge, this is the first diffusion-based framework to operate independently of discretization, offering a practical and flexible solution for forward and inverse problems in the context of PDEs. Code is available at https://github.com/neuraloperator/FunDPS.

## 1 Introduction

Conditional sampling is a ubiquitous task in machine learning and scientific computing that involves generating samples from a distribution conditioned on certain constraints or observations. This task appears naturally in many applications where we need to reconstruct high-dimensional data given partial, corrupted, or indirect measurements. It has been extensively studied in the image domain for tasks like inpainting, deblurring, and super-resolution [1–5].

In scientific domains, an example of conditional sampling is climate modeling, where scientists predict future atmospheric states based on limited sensor measurements. Traditional forecasting methods try to produce a single, best-case outcome based on the available data. However, the chaotic nature of weather systems means small uncertainties in initial measurements can lead to drastically different outcomes [6]. Conditional sampling methods can generate multiple plausible weather states consistent with the available data [7]. This probabilistic approach is especially valuable for predicting extreme weather events and making informed policy decisions. Beyond weather prediction,

---

[*]Equal contribution.
[†]Now at the University of Oxford.

39th Conference on Neural Information Processing Systems (NeurIPS 2025).

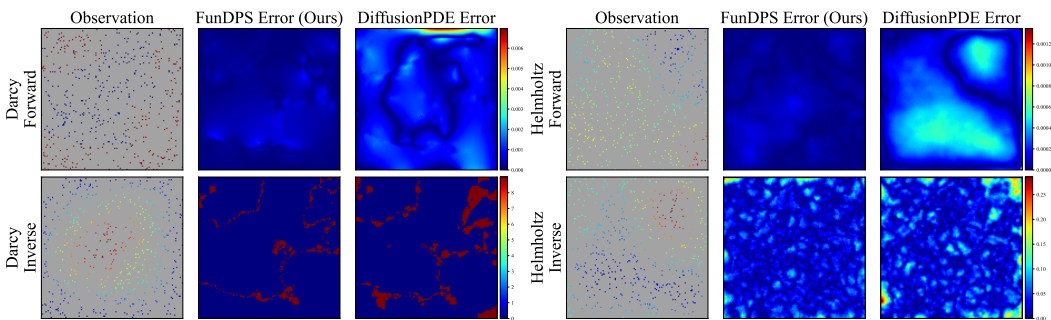

Figure 1: Comparison between our method (FunDPS) with the state-of-the-art diffusion baseline, DiffusionPDE, on the Darcy Flow and Helmholtz problems. The left column shows the sparse observation measurements (3% of total points), while the other two columns display the absolute reconstruction error of our method and DiffusionPDE, respectively. FunDPS achieves superior accuracy with an order of magnitude fewer sampling steps.

conditional sampling naturally arises in scientific tasks such as solving inverse problems in physical systems. These examples include recovering permeability fields in subsurface flows [8], inferring material properties in elasticity [9], and identifying initial conditions for fluid simulations [10].

Traditional numerical methods often struggle with ill-posed problems such as when observations are sparse or noisy, as they cannot effectively leverage prior knowledge about the solutions. While Markov Chain Monte Carlo (MCMC) methods can theoretically sample from the posterior distribution, they are computationally intensive and often require exponentially many iterations to converge, making them impractical for high-dimensional problems [11]. Moreover, constructing effective proposal distributions for MCMC is particularly challenging, leading to poor mixing times [12].

**Bayesian approach to conditional sampling**    A principled approach to conditional sampling is from a Bayesian perspective [13]. We aim to sample from the posterior distribution $p(\boldsymbol{a}|\boldsymbol{u}) \sim p(\boldsymbol{a})p(\boldsymbol{u}|\boldsymbol{a})$, where $\boldsymbol{u}$ represents our observations. In inverse problems, this posterior is defined through a forward operator $\boldsymbol{A}$ and the likelihood function $p(\boldsymbol{u}|\boldsymbol{a}) = p(\boldsymbol{u}|\boldsymbol{A}(\boldsymbol{a}))$. The forward operator $\boldsymbol{A}$ can take many forms depending on the application. For instance, when $\boldsymbol{A}$ is a masking operator that selects specific coordinates, it forms the reconstruction problem from sparse observations. When $\boldsymbol{A}$ represents a PDE solution operator mapping from parameters to solutions, we obtain parameter inference problems in physics. The likelihood function can also incorporate different noise assumptions to handle varying measurement uncertainties. This general framework effectively consolidates many conditional sampling problems into one unified mathematical formulation.

**Diffusion Models in Function Spaces**    Physical systems are inherently described by continuous functions rather than discrete grids. Consequently, methods developed for finite-dimensional settings tend to perform poorly when applied to different discretizations of infinite-dimensional problems. While diffusion models have been extended to function space settings, existing approaches have limitations. Denoising Diffusion Operator [14] uplifts the generative process to infinite dimensions by a function-valued annealed Langevin dynamics, but is limited to an uncontrollable generation pipeline. Baldassari et al. [15] suggests a conditional denoising estimator in infinite dimensions, yet its adaptability is limited by the requirement of a pre-trained conditional score model. In real-world applications, the density and configuration of sensors can vary significantly, and training a separate model for each measurement setup is too costly. This highlights the need for a method capable of performing diffusion posterior sampling in function spaces using an unconditional score model. The ability to use one such model for various downstream tasks offers substantial flexibility, effectively decoupling the core model development from specific application requirements.

**Contributions**    We introduce **Fun**ction-space **D**iffusion **P**osterior **S**ampling (FunDPS), a novel framework that leverages diffusion operators to address inverse problems by resolution-independent conditional sampling. Our contributions can be regarded as tri-fold:

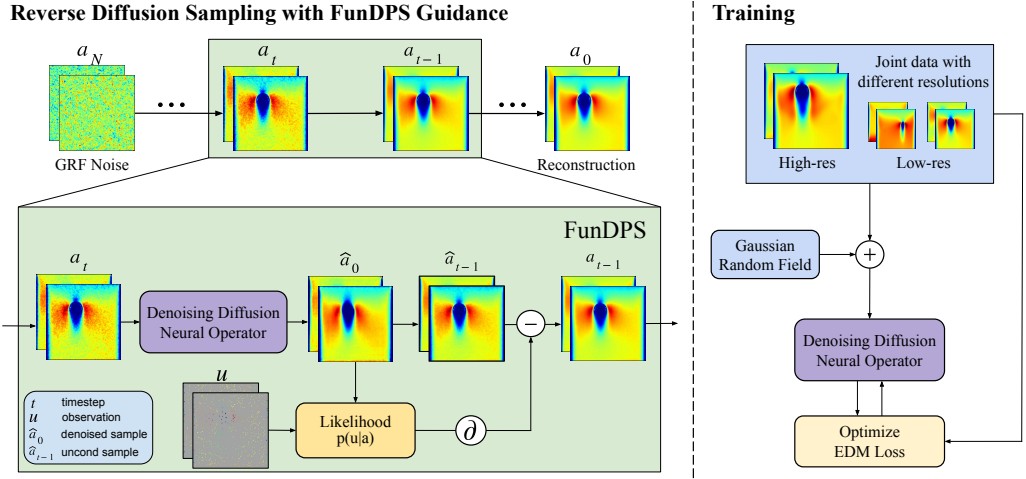

Figure 2: The sampling and training pipelines of FunDPS. During inference, we utilize a standard reverse diffusion approach with additional FunDPS guidance to drag the samples to the posterior. During training, the model is based on a U-shaped neural operator, and Gaussian random fields are used as the noise sampler to ensure consistency within function spaces. Notations are detailed in blue box in the bottom-left corner, where function $a$ jointly represents the PDE parameters and solution.

*1) Infinite-dimensional Tweedie's formula.*
We develop a novel theoretical framework for posterior sampling in infinite-dimensional spaces by extending Tweedie's formula to the Banach space setting. Tweedie's formula gives a closed-form expression for the posterior mean of the reverse diffusion process and serves as the basis for many diffusion solvers. However, prior to this work, it has only been shown to hold in *finite* dimensions. We rigorously establish a generalization of Tweedie's formula to *function spaces*, thereby providing a plug-and-play approach for guided sampling in infinite-dimensional inverse problems. Concretely, we study the scenario where a given data measure is perturbed via an additive Gaussian measure, and show an equivalence between the score of the resulting noisy distribution and the conditional expectation of the noise-free sample given a noisy one. This generalization, combined with a measure-theoretic decomposition of the conditional score, enables us to inject measurement consistency (or PDE constraints) into an *unconditional* function-space diffusion model at inference time.

*2) Multi-resolution conditional generation pipeline in function space (FunDPS).*
We propose the function-space diffusion posterior sampling method that builds on the above theoretical framework, the pipeline of which is shown in Figure 2. An unconditional diffusion model is first pretrained using the score matching objective as defined in [14]. The model works on a joint functional representation of PDE parameters and solutions, which allows us to solve forward problems, inverse problems, and a combination thereof using partial observations. The model employs a **multi-resolution training strategy**, which begins training on a coarse grid, then fine-tunes on a finer one. It **reduces GPU training hours by 25%**, which is enabled by the Gaussian random field (GRF)-based noise model and neural operator architecture. After training the unconditional model, we generate conditional samples as follows: The sample is initialized with GRF noise, followed by iterative denoising via a second-order deterministic sampler [16]. At each step, the sample evolves according to the conditional score operator, which decomposes into data prior (from the pretrained model) and observation-based likelihoods. Our generalized Tweedie's formula enables us to efficiently approximate the measurement log-likelihood. FunDPS can also employ **multi-resolution inference** by performing most sampling steps at a lower resolution and only upsampling towards the end, which alone **yields 2x speedup while maintaining accuracy**.

*3) Extensive experiments with SotA performance in both speed and accuracy.*
We test our approach on five challenging PDEs, which vary widely in their difficulty due to complex input distributions (GRF's multi-scale features), different boundary conditions (Dirichlet and periodic), and nonlinear patterns (Navier-Stokes). We simulate extreme obfuscation in measurements by masking. Specifically, for forward tasks (e.g., recovering the final state), we only observe 3% of the initial condition and reconstruct the entire solution, while for inverse tasks, we observe

only 3% of the final state and reconstruct unknown initial or coefficient fields. Despite these difficulties, our method consistently achieves top performance (see Figure 1). Across all five tasks, we **reduce the average error by roughly 32%** compared to the state-of-the-art diffusion-based solver (DiffusionPDE [17], which operates at a fixed trained resolution) and surpass deterministic neural PDE solvers by even larger margins. Moreover, our reverse diffusion process requires only 200–500 steps to converge—up to $10\times$ **fewer** than DiffusionPDE—while maintaining superior accuracy (see Figure 3a). We attribute these improvements primarily to FunDPS's function-space formulation, which aligns better with physics functions and applies smoother guidance, **cutting inference wall-clock time by $25\times$** compared to DiffusionPDE with minimal impact on accuracy.

## 2    Related Works

**Neural PDE solvers**    Neural PDE solvers aim to approximate PDE solutions or operators using deep networks. Physics-informed neural networks (PINNs) [18] incorporate PDE residuals and boundary conditions into the loss function, which enables both forward and inverse problem solving but often at the cost of challenging optimization or limited scalability [19, 20]. Operator-learning approaches such as Fourier Neural Operator (FNO) [21] and DeepONet [22, 23] learn resolution-invariant mappings between function spaces. Meanwhile, graph-based neural PDE solvers [24, 25] treat spatial discretization as a graph rather than functions and apply message passing to approximate PDE dynamics on potentially irregular domains. Despite these breakthroughs, most existing works yield single (deterministic) outputs rather than sampling from a posterior distribution over PDE equations in function spaces, an aspect that we address through our diffusion-based approach.

**Inverse PDE problems**    Inverse problems in PDEs, like recovering material properties or flow states from sparse or noisy measurements, have been mainly tackled by PDE-constrained optimization or Bayesian inference settings. Recently, data-driven approaches like PINN, FNO, and DeepONet [18, 21, 22] have emerged as efficient approaches to tackle PDE problems. However, their main target is not recovering the full fields from partial observations, but learning the function-space mappings. More recently, Energy Transformer [26] can reconstruct full fields from incomplete or irregular data, but they are limited by the patch size. Others have explored cGANO for broadband ground-motion synthesis [27], unified latent representations for forward–inverse subsurface imaging [28], measurement-guided diffusion in geophysical tasks [29], and physically consistent score-based methods with PDE constraints [30]. Although these works demonstrate promising directions and partial overlap with our goals, they typically rely on discretized image domains or separately trained physical models, lacking resolution-invariant features and flexibility.

**Diffusion-based posterior sampling**    Diffusion models have demonstrated high quality and stability in generation tasks [31–33, 16]. By learning a score function—a vector field pointing towards high-density data regions at different noise levels—we can reverse the noising process to sample from the prior distribution $\nu(\boldsymbol{a})$ [34–36]. Diffusion models have been widely used to solve inverse problems in a plug-and-play manner, utilizing a pre-trained unconditional diffusion model as a prior and its sampling process to integrate constraints [37, 38]. One key advantage of these methods is their flexibility—the same unconditional model can be used for various downstream tasks without retraining. Various approaches have been proposed to leverage diffusion priors, ranging from guidance terms or resampling strategies within the generative process [39, 40] to integrations within variational frameworks [41, 42]. While widely researched for fixed-resolution inverse problems like images, their application to partial differential equations (PDEs) in function spaces is less explored. DiffusionPDE [17] learns joint distributions of PDE parameters and solutions, but its reliance on fixed discretizations limits its practical applicability in scientific computing.

**Generative neural operators**    For physics-informed problems, developing true *function-space* generative methods is crucial as data are inherently functions. Lim et al. [14] proposes Denoising Diffusion Operators (DDOs), which extend diffusion processes to Gaussian random fields and prove resolution-independence under discretization, while Pidstrigach et al. [43] provides a rigorous mathematical framework for infinite-dimensional diffusion models that preserve key properties as discretization is refined. Another work uses *adversarial* training on function spaces [44], which introduces Generative Adversarial Neural Operators (GANO) to learn push-forward maps between probability measures in infinite-dimensional Hilbert spaces. Flow matching techniques [45] offer an

alternative by directly learning a velocity field whose induced transport matches the target distribution. This framework has very recently been extended to infinite-dimensional settings [46–48]. Kerrigan et al. [49] further applies functional flow matching in a conditional inference context, providing a rigorous training scheme for downstream tasks. In contrast, we leverage a pre-trained functional prior and simply compose it with arbitrary measurement operators at test time in a plug-and-play manner, without requiring task-specific re-training.

# 3    Method

We now present our function-space diffusion posterior sampling (FunDPS) framework. We begin by formulating the inverse problem in function spaces and establishing the Bayesian framework. We then derive the conditional score through the decomposition of likelihood and prior terms and develop the necessary approximations for practical implementation. Finally, we describe the complete FunDPS algorithm, including our multi-resolution training strategy.

## 3.1    Problem Settings

**PDE-based inverse problems**    We focus on general inverse problems in function spaces, with the goal of retrieving an unknown function $\boldsymbol{a} \in \mathcal{A}$ from a measurement function $\boldsymbol{u} \in \mathcal{U}$ given by

$$\boldsymbol{u} = \boldsymbol{A}(\boldsymbol{a}) + \varepsilon \quad \text{with} \quad \boldsymbol{A} : \mathcal{A} \to \mathcal{U}, \tag{1}$$

where $\varepsilon$ is an $\mathcal{U}$-valued random variable representing the measurement noise and $\mathcal{A}$ and $\mathcal{U}$ are separable Hilbert spaces. Such inverse problems frequently appear in the context of PDEs

$$L_c f = 0 \text{ (on } D) \quad \text{and} \quad f|_{\partial D} = g, \tag{2}$$

where $L_c$ is a differential operator depending on a coefficient function $c$, the function $g$ encodes boundary or initial values on the domain $D$, and $f$ is the solution[3] function. For classical inverse problems, $\boldsymbol{A}$ can be the operator mapping the parameter functions of a PDE, i.e. coefficient functions $c$ and boundary values $g$, to the corresponding solution $f$. Additionally, $\boldsymbol{A}$ can combine both a PDE solution operator and a masking operator for solving inverse problems with sparse measurements.

**Bayesian perspective**    For many practical solutions and masking operators, the inverse problem is ill-posed, i.e., $\boldsymbol{A}$ is not injective nor stable, meaning that small changes in $\boldsymbol{u}$ can cause large variations in $\boldsymbol{a}$. To address these issues, we consider a Bayesian perspective [13], where we assume a prior distribution $\nu(\boldsymbol{a})$ on $\mathcal{A}$ with the aim of sampling from the corresponding posterior distribution $\nu^{\boldsymbol{u}} = \nu(\boldsymbol{a}|\boldsymbol{u})$. All distributions, including the prior $\nu$ and posterior $\nu^{\boldsymbol{u}}$, are considered as probability measures, and the existence of densities in infinite-dimensional spaces often requires careful treatment.

We assume that the measurement noise $\varepsilon$ is a Gaussian random field (GRF) with covariance operator $\mathbf{C}_\eta$, i.e., $\varepsilon$ is drawn from the Gaussian measure $\eta = \mathcal{N}(0, \mathbf{C}_\eta)$ on $\mathcal{U}$, which is the natural extension of Gaussian noise to function spaces [50]. Observe that, given $\boldsymbol{a}$, we have that $\boldsymbol{u} \mid \boldsymbol{a} \sim \eta^{\boldsymbol{A}(\boldsymbol{a})} = \mathcal{N}(\boldsymbol{A}(\boldsymbol{a}), \mathbf{C}_\eta)$ is drawn from a Gaussian measure $\eta^{\boldsymbol{A}(\boldsymbol{a})}$ with mean $\boldsymbol{A}(\boldsymbol{a})$. When the noise-free observation $\boldsymbol{A}(\boldsymbol{a})$ is an element of the Cameron-Martin space $\mathbf{C}_\eta^{1/2}(\mathcal{U}) =: H(\eta) \subset \mathcal{U}$ associated with $\mathbf{C}_\eta$, Cameron-Martin theorem allows us to compute the Radon-Nikodym derivative [4]

$$\frac{\mathrm{d}\eta^{\boldsymbol{A}(\boldsymbol{a})}}{\mathrm{d}\eta}(\boldsymbol{u}) = \exp\left(\langle \boldsymbol{u}, \boldsymbol{A}(\boldsymbol{a})\rangle_{H(\eta)} - \tfrac{1}{2}\|\boldsymbol{A}(\boldsymbol{a})\|_{H(\eta)}^2\right). \tag{3}$$

In other words, Eq. (3) gives the density of $\eta^{\boldsymbol{a}}$ with respect to the noise measure $\eta$. We will use $\Phi : \mathcal{A} \times \mathcal{U} \to \mathbb{R}$ to represent the function $\Phi(\boldsymbol{a}, \boldsymbol{u}) = \langle \boldsymbol{u}, \boldsymbol{A}(\boldsymbol{a})\rangle_{\mathcal{U}_0} - \tfrac{1}{2}\|\boldsymbol{A}(\boldsymbol{a})\|_{\mathcal{U}_0}^2$. Loosely speaking, $\Phi(\boldsymbol{a}, \boldsymbol{u})$ is the log-likelihood of $\boldsymbol{u}$ given $\boldsymbol{a}$. In the special case that $\boldsymbol{u} \in \mathbb{R}^n$ is finite dimensional (for instance, when $\boldsymbol{A}$ is the composition of a PDE solution operator and a finite observation mask), the measure $\eta$ corresponds to a mean-zero Gaussian random variable with covariance matrix $\mathbf{C}_\eta \in \mathbb{R}^{n \times n}$. In this case, when $\mathbf{C}_\eta$ has full rank, the Cameron-Martin space is $\mathbf{C}_\eta^{1/2}(\mathbb{R}^n) = \mathbb{R}^n$ so that the log-likelihood is defined for any value of $\boldsymbol{A}(\boldsymbol{a})$.

---

[3]We assume that the considered PDEs allow for unique, strong solutions in a suitable space of functions.

[4]We recall that the inner product and norm on the Cameron-Martin space $\mathcal{U}_0$ are readily computed via

$$\langle \boldsymbol{u}_0, \boldsymbol{u}_1\rangle_{H(\eta)} = \langle \mathbf{C}_\eta^{-1/2}\boldsymbol{u}_0, \mathbf{C}_\eta^{-1/2}\boldsymbol{u}_1\rangle_{\mathcal{U}}, \qquad \boldsymbol{u}_0, \boldsymbol{u}_1 \in H(\eta).$$

By Bayes' rule [13], the posterior $\nu^{\boldsymbol{u}}$ is absolutely continuous with respect to the prior measure $\nu$, and moreover, the corresponding Radon-Nikodym derivative is proportional to the likelihood, i.e., $\frac{d\nu^{\boldsymbol{u}}}{d\nu}(\boldsymbol{a}) \propto \exp\left(\Phi(\boldsymbol{a}, \boldsymbol{u})\right)$ with the constant of proportionality depending on $\boldsymbol{u}$ but not $\boldsymbol{a}$.

**Diffusion prior**    We propose to learn the prior distribution $\nu(\boldsymbol{a})$ from data using the recent extension of diffusion models to function spaces [14, 51]. These methods define a sequence of distributions $\nu_t(\boldsymbol{a}_t)$ that progressively add noise to the data until approximately reaching a tractable latent distribution $\mu_T \approx \boldsymbol{\Gamma}$. Learning a score approximator $\boldsymbol{D}_\theta(\boldsymbol{a}_t, t) \approx \mathbb{E}[\boldsymbol{a}_0 | \boldsymbol{a}_t]$ from the data via a variant of score matching, one can approximately reverse the noising process[5] and sample from the data distribution. Moreover, by replacing the score with a conditional expectation, one can solve the reverse SDE given initial conditions [43]. Notably, the denoiser $\boldsymbol{D}_\theta$ is parametrized as a neural operator [14, 52], and the noise is chosen to be Gaussian random fields, as opposed to neural networks and multivariate Gaussian random variables for the finite-dimensional setting (see Appendix A for more details). The diffusion prior will be updated by (approximated) conditional likelihood during inference in order to solve inverse problems.

## 3.2    Conditional Score via Likelihood and Prior Score

In our setting, the unconditional forward process of the diffusion model may be simulated via

$$\boldsymbol{a}_t = \boldsymbol{a}_0 + \sigma_t \varepsilon_t \qquad\qquad \varepsilon_t \sim \gamma, \ \boldsymbol{a}_0 \sim \nu \qquad\qquad (4)$$

for some specified time-dependent constants $\sigma_t > 0$ [14]. Here, $\gamma = \mathcal{N}(0, \mathbf{C}_\gamma)$ is now a Gaussian measure on the space $\mathcal{A}$. We will also use $\gamma_t = \mathcal{N}(0, \sigma_t^2 \mathbf{C}_\gamma)$ to represent the corresponding Gaussian measure with scaled covariance. The spaces $H(\gamma), H(\gamma_t)$ represent the Cameron-Martin spaces associated with these Gaussian measures. We will write $\nu_0(\boldsymbol{a}_0) = \nu(\boldsymbol{a})$ for the prior distribution over $\boldsymbol{a}$ and $\nu_t(\boldsymbol{a}_t)$ for the marginal distribution of $\boldsymbol{a}_t$ obtained during the forward process. Similarly, the *conditional* forward process is obtained in the same manner as Eq. (4) except with initial conditions $\boldsymbol{a}_0 \sim \nu^{\boldsymbol{u}}$ drawn from the posterior corresponding to a given, fixed $\boldsymbol{u}$. We will use $\nu_0^{\boldsymbol{u}} = \nu^{\boldsymbol{u}}$ and $\nu_t^{\boldsymbol{u}}$ for the corresponding measures. Informally, $\nu_t^{\boldsymbol{u}}$ can be thought of as $\nabla_{\boldsymbol{a}} p_t(\boldsymbol{a}_t | u)$.

As in prior work [14, 43], we make the assumption that the prior $\nu_0$ is supported on $H(\gamma)$, i.e., $\nu(H(\gamma)) = 1$. Note that as the posterior $\nu_0^{\boldsymbol{u}} \ll \nu_0$ is absolutely continuous with respect to the prior, we also have that $\nu_0^{\boldsymbol{u}}(H(\gamma)) = 1$. Under this assumption, both $\nu_t$ and $\nu_t^{\boldsymbol{u}}$ are equivalent to $\gamma_t$ in the sense of mutual absolute continuity (see Appendix C and [14, Lemma 13]).

In this case, by the Radon-Nikodym theorem there exists a density $d\nu_t^{\boldsymbol{u}} / d\gamma_t$. We assume that the logarithm of this density is Fréchet differentiable along $H(\gamma_t)$. The object of interest when seeking to sample from the posterior $\nu^{\boldsymbol{u}}$ using a diffusion model is the *conditional score* of $\nu_t$, defined as

$$D_{H(\gamma_t)} \log \frac{d\nu_t^{\boldsymbol{u}}}{d\gamma_t} : \mathcal{A} \to H(\gamma_t)^*. \qquad\qquad (5)$$

The unconditional score of $\nu_t$ is defined analogously. Previous work [14] uses this notion of a score to build function-space diffusion models. To simplify the notation, we write $\nabla_{\boldsymbol{a}_t} = D_{H(\gamma_t)}$ as shorthand for this Fréchet derivative.

However, in our setup, we do not assume we have access to the score of $\nu_t^{\boldsymbol{u}}$, but rather only to the unconditional score of $\nu_t$ and the log-likelihood function $\Phi(\boldsymbol{a}_0, \boldsymbol{u})$. To overcome this, note that

$$\nabla_{\boldsymbol{a}_t} \log \frac{d\nu_t^{\boldsymbol{u}}}{d\gamma_t}(\boldsymbol{a}_t) = \nabla_{\boldsymbol{a}_t} \log \frac{d\nu_t^{\boldsymbol{u}}}{d\nu_t}(\boldsymbol{a}_t) + \nabla_{\boldsymbol{a}_t} \log \frac{d\nu_t}{d\gamma_t}(\boldsymbol{a}_t) = \nabla_{\boldsymbol{a}_t} \tilde{\Phi}_t(\boldsymbol{a}_t, \boldsymbol{u}) + \nabla_{\boldsymbol{a}_t} \log \frac{d\nu_t}{d\gamma_t}(\boldsymbol{a}_t), \quad (6)$$

where $\tilde{\Phi}_t(\boldsymbol{a}_t, \boldsymbol{u}) = \log((d\nu_t^{\boldsymbol{u}} / d\nu_t)(\boldsymbol{a}_t))$ is the log-likelihood of $\boldsymbol{u} \mid \boldsymbol{a}_t$. Thus, we have managed to decompose the conditional score into a sum of a likelihood term and the prior score as desired.

## 3.3    Approximating the Likelihood in Function Spaces

While the calculations in the previous section in principle allow for plug-and-play posterior sampling, a major difficulty is that the time-dependent likelihood $\tilde{\Phi}_t$ is intractable. In particular, from Eq. (3) we know the log-likelihood of $\boldsymbol{u}$ given a noise-free sample $\boldsymbol{a}_0$, but Eq. (6) requires us to have access

---

[5]Following common conventions, we will also refer to the noising process as *forward process*.

to the log-likelihood of $\boldsymbol{u}$ given a *noisy* sample $\boldsymbol{a}_t$. In the next two sections, we develop a method for approximating this likelihood through a conditional expectation.

Let us write $\mu_t^{\boldsymbol{a}_t}$ for the conditional distribution of $\boldsymbol{u} \mid \boldsymbol{a}_t$. This measure can be sampled from by first predicting the clean $\boldsymbol{a}_0$ from the noisy $\boldsymbol{a}_t$, followed by sampling from the noise measure $\eta^{\boldsymbol{A}(\boldsymbol{a}_0)}$ according to Eq. (1). Hence, for any measurable $U \subseteq \mathcal{U}$, we have

$$\mu_t^{\boldsymbol{a}_t}(U) = \int_{\mathcal{A}} \eta^{\boldsymbol{A}(\boldsymbol{a}_0)}(U) \, \mathrm{d}\nu_{0|t}^{\boldsymbol{a}_t}(\boldsymbol{a}_0) \tag{7}$$

where $\nu_{0|t}^{\boldsymbol{a}_t}$ is the conditional measure of the reverse diffusion process given a noisy $\boldsymbol{a}_t$. Denote by $\hat{\boldsymbol{a}}_0(\boldsymbol{a}_t) = \mathbb{E}[\boldsymbol{a}_0 \mid \boldsymbol{a}_t]$ the expected value of this measure. (We will sometimes write $\hat{\boldsymbol{a}}_0$ to simplify the notation when the dependency on $\boldsymbol{a}_t$ is clear from context.) Approximating $\nu_{0|t}^{\boldsymbol{a}_t} \approx \delta[\hat{\boldsymbol{a}}_0]$ with its mean, under the assumption that $\boldsymbol{A}(\boldsymbol{a}_0) \in H(\eta)$ is an element of the CM space of $\eta$, we have

$$\mu_t^{\boldsymbol{a}_t}(U) = \int_{\mathcal{A}} \eta^{\boldsymbol{A}(\boldsymbol{a}_0)}(U) \, \mathrm{d}\nu_{0|t}^{\boldsymbol{a}_t}(\boldsymbol{a}_0) = \int_{\mathcal{A}} \int_U \frac{\mathrm{d}\eta^{\boldsymbol{a}_0}}{\mathrm{d}\eta}(\boldsymbol{u}) \, \mathrm{d}\nu_{0|t}^{\boldsymbol{a}_t}(\boldsymbol{a}_0) \approx \int_U \frac{\mathrm{d}\eta^{\boldsymbol{A}(\hat{\boldsymbol{a}}_0)}}{\mathrm{d}\eta}(\boldsymbol{u}) \, \mathrm{d}\eta(\boldsymbol{u}). \tag{8}$$

Note further that Eq. (7) shows that if the prior $\nu_0$ is such that $\boldsymbol{A}(\boldsymbol{a}_0) \in H(\eta)$ almost surely, then $\mu_t^{\boldsymbol{a}_t} \ll \eta$. In all, we have shown

$$\frac{\mathrm{d}\mu_t^{\boldsymbol{a}_t}}{\mathrm{d}\eta}(\boldsymbol{u}) \approx \frac{\mathrm{d}\eta^{\boldsymbol{A}(\hat{\boldsymbol{a}}_0)}}{\mathrm{d}\eta}(\boldsymbol{u}). \tag{9}$$

Carrying this approximation over into the log-likelihood, we obtain $\tilde{\Phi}_t(\boldsymbol{a}_t, \boldsymbol{u}) \approx \Phi(\hat{\boldsymbol{a}}_0, \boldsymbol{u})$ as defined by Eq. (3). Thus, when we are able to calculate $\nabla_{\boldsymbol{a}_t} \Phi(\hat{\boldsymbol{a}_0}(\boldsymbol{a}_t), \boldsymbol{u})$, we may substitute this expression into Eq. (6) in order to guide the diffusion process to approximately sample from the posterior $\nu^{\boldsymbol{u}}$.

In the special case that $\mathcal{U}$ is finite dimensional (e.g., when $\boldsymbol{A}$ represents observing our PDE at finitely many locations) and $C_\eta$ is full rank, observe that via a straightforward calculation we have

$$\Phi(\hat{\boldsymbol{a}}_0, \boldsymbol{u}) = -\tfrac{1}{2} \left( \|\boldsymbol{u} - \boldsymbol{A}\hat{\boldsymbol{a}}_0\|_{H(\eta)}^2 - \|\boldsymbol{u}\|_{H(\eta)}^2 \right) \tag{10}$$

and so in this case we are justified in using an approximation to our log-likelihood of the form

$$\nabla_{\boldsymbol{a}_t} \tilde{\Phi}_t(\boldsymbol{a}_t, \boldsymbol{u}) \approx \nabla_{\boldsymbol{a}_t} \left( -\tfrac{1}{2} \|\mathbf{C}_\eta^{-1/2}(\boldsymbol{u} - \boldsymbol{A}\hat{\boldsymbol{a}}_0(\boldsymbol{a}_t))\|_{\mathcal{U}}^2 \right) \tag{11}$$

as the two differ by a constant depending only on $\boldsymbol{u}$. However, when $\mathcal{U}$ is infinite dimensional, the Cameron-Martin space $H(\eta)$ has measure zero under $\eta$, and so $\boldsymbol{u} - \boldsymbol{A}(\hat{\boldsymbol{a}_0})$ almost surely not be an element of $H(\eta)$ in which case Eq. (10) is ill-defined. It is also worth noting that this approximation is less accurate when noise level is high [53], which opens future work in this topic.

### 3.4 Approximation of Conditional Expectation via Tweedie's Formula

Tweedie's formula [54] plays a fundamental role in diffusion solvers by providing a link between score functions and conditional expectations. It is a cornerstone in many guided sampling methods, including MCG [38], DPS [40], and PSLD [55]. In our framework, the approximated likelihood also relies on this efficient estimation of conditional expectations $\mathbb{E}[\boldsymbol{a}_0 \mid \boldsymbol{a}_t]$, but in function spaces.

Generalization of Tweedie's formula to function spaces requires careful treatment of several concepts. In infinite-dimensional spaces, we can no longer rely on probability density functions or standard gradients as in the finite-dimensional case. The notion of conditional expectation must be handled through measure-theoretic tools, and the score function needs to be redefined using the Fréchet derivative and the Riesz representation theorem. Here we present our extension below.

**Theorem 3.1** (Tweedie's formula in infinite-dimensional Banach spaces). *Let $B$ be a separable Banach space. Assume that $\mu(H(\gamma)) = 1$, the score of $\nu$ is Fréchet differentiable along $H(\gamma)$, and that the Fréchet derivatives of $\mathrm{d}\gamma^x / \mathrm{d}\gamma$ are $\mu$-almost surely bounded by a $\mu$-integrable function. Then, for $\nu$-almost every $y$,*

$$\mathbb{E}[X \mid Y = y] = R \left( D_{H(\gamma)} \log \frac{\mathrm{d}\nu}{\mathrm{d}\gamma}(y) \right). \tag{12}$$

**Proof.** See Appendix C. $\qquad\square$

### 3.5 FunDPS: Function-Space Diffusion Posterior Sampling

With the analytically tractable likelihood approximation, we can now proceed to the posterior sampling step. We will introduce the guidance term for the reverse diffusion process by defining a differentiable likelihood term given a measurement function. The FunDPS algorithm follows, with special attention to some design details of our implementation.

**Guidance**   With the extended Tweedie's formula, we now have a closed form representation of $\hat{\boldsymbol{a}}_0(\boldsymbol{a}_t) = \mathbb{E}[\boldsymbol{a}_0 \mid \boldsymbol{a}_t]$ given by Theorem 3.1. It becomes differentiable when we use a neural operator to simulate the score function, enabling us to compute the likelihood function given in Eq. (11).

In practice, our observations $\boldsymbol{u}$ will be finite dimensional. We also assume that the covariance $C_\eta$ of the observation noise is approximately uncorrelated, such that $C_\eta^{-1}$ can be well-approximated by a scaled version $C_\eta^{-1} \approx cI$ of the identity. Plugging this and Eq. (11) into Eq. (6) with the trained score operator $\boldsymbol{D}_\theta$, we can now formalize the conditional score operator, acting as the guidance for posterior sampling

$$\nabla_{\boldsymbol{a}_t} \log \frac{\mathrm{d}\nu_t^{\boldsymbol{u}}}{\mathrm{d}\gamma_t}(\boldsymbol{a}_t) = \nabla_{\boldsymbol{a}_t} \log \frac{\mathrm{d}\nu_t}{\mathrm{d}\gamma_t}(\boldsymbol{a}_t) + \nabla_{\boldsymbol{a}_t} \tilde{\Phi}_t(\boldsymbol{a}_t, \boldsymbol{u}) \approx \boldsymbol{D}_\theta(\boldsymbol{a}_t, t) - \tfrac{c}{2} \nabla_{\boldsymbol{a}_t} \|\boldsymbol{u} - \boldsymbol{A}(\hat{\boldsymbol{a}}_0)\|_{\mathcal{U}}^2. \quad (13)$$

When there are multiple types of observation, we further define the total log likelihood as a weighted sum of individual observation log likelihoods. This formulation introduces a vector of hyperparameters $\boldsymbol{\zeta}$, where each component scales its corresponding likelihood term. These weights can be interpreted as confidence measures for observations. For simplicity, we also absorb $\frac{c}{2}$ into $\boldsymbol{\zeta}$.

Algorithm 2 depicts this reverse diffusion process, where we iteratively update the samples $\boldsymbol{a}_i$ using the consistency between the given measurement $\boldsymbol{u}$ and the one obtained from the denoiser, i.e., $\boldsymbol{A}(\boldsymbol{D}_\theta(\boldsymbol{a}_i, t_i))$. Specifically, we propose the update rule

$$\boldsymbol{a}_{i+1} \leftarrow \boldsymbol{a}_i - \boldsymbol{\zeta} \cdot \nabla_{\boldsymbol{a}_i} \|\boldsymbol{u} - \boldsymbol{A}(\boldsymbol{D}_\theta(\boldsymbol{a}_i, t_i))\|_{\mathcal{U}}^2, \quad (14)$$

where $\boldsymbol{\zeta}$ is a predefined guidance weight as above.

**Joint embedding**   In practice, the solution and the parameter functions are typically measured by sensors and can only be partially observed [56]. Hence, we focus on the general case where the operator $\boldsymbol{A}$ is an arbitrary masking operator, specifying the coordinates of the sensors, and the function $\boldsymbol{a}$ *jointly* represents the PDE parameters and solution. Our goal is learning the prior distribution of both initial and solution states via diffusion models. We then recover classical inverse problems by fully masking the coefficients. On the other hand, forward problems correspond to reconstructing the masked full solution from partial observations of the parameters. By partially masking either channel, sparse reconstruction problems are resolved.

**Multi-resolution training**   Inspired by other works on operator learning [57, 58], we introduce a new training technique to learn the prior distribution with reduced computational costs. We first train the diffusion model on low-resolution data for a majority of the epochs and only train on higher resolution for the final epochs. This curriculum learning approach guides the model to efficiently learn coarser information at the earlier stages of training and finer high-frequency details in the later stages. Due to the discretization invariance of neural operators, the resulting model exhibits similar performance as training only on high resolutions, almost at the cost of low-resolution training.

**Multi-resolution inference**   Beyond reducing steps, inference time per step can be accelerated by initially processing at a lower resolution and progressively increasing it. This aligns with the concept of diffusion models generating autoregressively in resolution or frequency space [59]. We propose ReNoise, a bi-level multi-resolution inference method. Initially, samples are processed at a lower resolution. Subsequently, they are upscaled, and inspired by ReSample[60], additional noise is introduced to mitigate artifacts and noise level mismatch arising from upscaling, followed by a few denoising steps at the target resolution. This enables us to achieve equivalent accuracy in half the computing time. We illustrate the pipeline in Figure 3b with implementation details in Appendix H.4.

## 4 Experiments

We will now showcase the efficiency, robustness, and multi-resolution generalizability of our proposed FunDPS through various tasks. Additional and ablation experiments can be found in Appendix H.

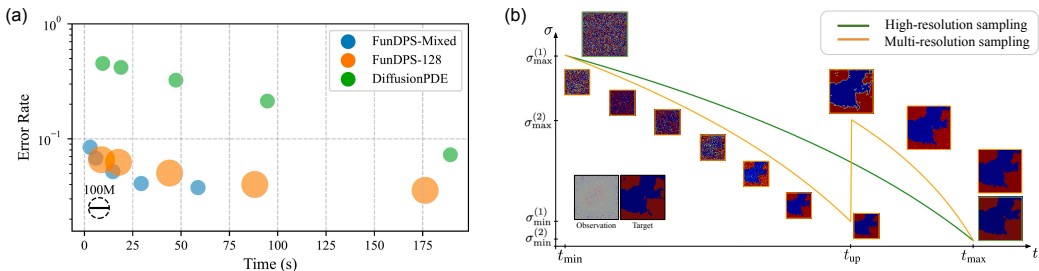

Figure 3: (a) Comparison of FunDPS and DiffusionPDE in terms of accuracy and inference time with varying step sizes; (b) Demonstration of multi-resolution inference pipeline. $\sigma$ is in log-scale.

## 4.1 Experimental Setup

**Datasets & Tasks**  We validate our approach by solving both forward and inverse problems on five different PDE problems. The problems span a range of difficulties, characterized by complex input distributions arising from the multi-scale features of GRFs, varying boundary conditions (Dirichlet and periodic), and nonlinear patterns inherent in the Navier-Stokes equations. We provide a detailed description of each PDE in Appendix E. The objective is to solve forward and inverse problems in sparse sensor settings. For forward tasks, we only observe 3% of the initial condition before reconstructing the entire solution. Conversely, for inverse tasks, we only observe 3% of the final state before reconstructing unknown initial or coefficient fields.

**Training & Inference**  We base our code on EDM-FS [61]. In particular, we adopt a U-shaped neural operator architecture [52] as the denoiser $D_\theta$ and modify the noising process according to the discussion above. Our denoiser has 54M parameters, which is similar to DiffusionPDE's network size. The implementation is detailed in Appendix F. The guidance weight $\zeta$ in Eq. (14) is tuned on a small validation set, provided in Appendix G.

**Baselines**  Baseline comparisons include well-known deterministic PDE solvers such as FNO [21], PINO [62], DeepONet [22] and PINN [18], as well as the state-of-the-art diffusion-based approach, DiffusionPDE [17]. We had correspondence with DiffusionPDE's authors about its reproducibility issues. We use our reproduced results for comparison. Please refer to the Appendix I for details.

## 4.2 Results

**Main results**  We evaluate our method on both forward and inverse problems across five PDE tasks with sparse observations. Results are shown in Table 1. Even with severe occlusions, where only 3% of function points are visible, we are able to reconstruct both states effectively. Our approach always achieves the best results among all the baselines with the lowest error rates, surpassing all baselines with 32% higher accuracy with one-fourth steps. We provide qualitative results in Figure 1 and Appendix K. We also compared all methods on classical fully-observed forward and inverse problems, where FunDPS also showed superior performance as in Table 6. Further, we test our framework with diverse guidance methods to show its adaptability (Figure 6 and Table 5). We also conducted an ablation study on the effectiveness of our design choices in Table 12. Additionally, we provide results by using FDM as forward operator and compare with joint learning in Appendix D.

**Inference speed**  We compare the inference time of our method with the diffusion-based solver, DiffusionPDE, as shown in Figure 3. With a 200-step discretization of the reverse-time SDE, our method achieved superior accuracy with only one-tenth of the integration steps compared to DiffPDE. When we increase the number of steps, FunDPS further reduces the error, which suggests that the scaling law of inference time may hold for solving PDE problems with guided diffusion. This highlights the effectiveness of our function space formulation and guidance mechanism.

Regarding the actual run time, our model averages 15s/sample for 500 steps (without multi-resolution inference technique) on a single NVIDIA RTX 4090 GPU, while DiffusionPDE takes 190s/sample for 2000 steps on the same hardware and the same $128 \times 128$ discretization. This superior performance

Table 1: **Comparison of different models on five PDE problems** (in $L^2$ relative error)

| | Steps ($N$) | Darcy Flow | | Poisson | | Helmholtz | | Navier-Stokes | | Navier-Stokes (BCs) | |
|---|---|---|---|---|---|---|---|---|---|---|---|
| | | Forward | Inverse | Forward | Inverse | Forward | Inverse | Forward | Inverse | Forward | Inverse |
| FunDPS (ours) | 200 | 2.88% | 6.78% | 2.04% | 24.04% | 2.20% | 20.07% | 3.99% | 9.87% | 5.91% | 4.31% |
| FunDPS (ours) | 500 | **2.49%** | **5.18%** | **1.99%** | **20.47%** | **2.13%** | **17.16%** | **3.32%** | **8.48%** | **4.90%** | **4.08%** |
| DiffusionPDE | 2000 | 6.07% | 7.87% | 4.88% | 21.10% | 12.64% | 19.07% | 3.78% | 9.63% | 9.69% | 4.18% |
| FNO | - | 28.2% | 49.3% | 100.9% | 232.7% | 98.2% | 218.2% | 101.4% | 96.0% | 82.8% | 69.6% |
| PINO | - | 35.2% | 49.2% | 107.1% | 231.9% | 106.5% | 216.9% | 101.4% | 96.0% | 81.1% | 69.5% |
| DeepONet | - | 38.3% | 41.1% | 155.5% | 105.8% | 123.1% | 132.8% | 103.2% | 97.2% | 97.7% | 91.9% |
| PINN | - | 48.8% | 59.7% | 128.1% | 130.0% | 142.3% | 160.0% | 142.7% | 146.8% | 100.1% | 105.5% |

can be attributed to two factors: our efficient implementation increases our steps per second, and we require significantly fewer steps than pixel-space models.

**Multi-resolution training**   For our main results, we employed a two-phase training strategy: training primarily on low-resolution data before switching to high resolution in the final epochs. As shown in Table 7, this approach achieves comparable accuracy to models trained solely on high-resolution grids, while requiring only 25% of the GPU hours.

**Multi-resolution inference**   Thanks to its multi-resolution nature, neural operators can be trained on multiple resolutions and applied to data of different resolutions. We can reduce significant inference time by performing most sampling steps at low resolution and upsampling only near the end to finalize high-frequency details. We found that upsampling in the middle of the diffusion process worked poorly, which we attribute to the difficulty of preserving GRF properties during upscaling. Hence, we propose a bi-level sampling process, ReNoise, that mitigates the upsampling artifacts by adding noise for improved correction potential. The implementation details are given in Appendix H.4.

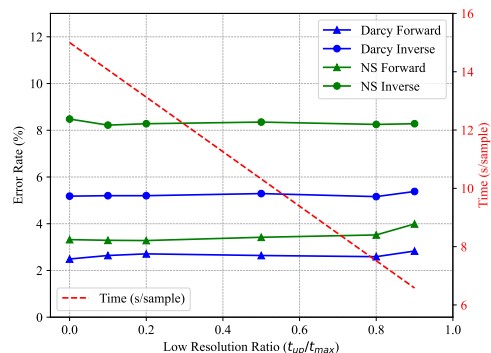

Figure 4: Comparison of the accuracy of FunDPS with ReNoise under different ratios of low-resolution inference steps.

We provide the multi-resolution inference results in Figure 4. With 80% of steps performed at low resolution, ReNoise sustains similar accuracy, yielding a further 2x speed improvement to 7.5s/sample–25 times faster than DiffusionPDE.

# 5   Conclusion

We introduce a novel discretization-agnostic generative framework for solving inverse problems in function spaces. Our framework supports sampling from posterior distributions with generalizability across different resolutions. We provide theoretical foundations for our framework with the extension of Tweedie's formulation to function spaces. We verified our approach with various settings and PDEs, achieving 32% higher accuracy than baselines while reducing time by an order of magnitude.

**Limitations**   Incorporating PDE loss in FunDPS yields only marginal improvements over pure observation-based guidance. We attribute this partially to numerical errors introduced by finite difference approximations of PDE operators. Second, our approach still requires manual tuning of guidance weights for different problems. Future work could explore improved techniques to better preserve the continuous nature of the PDEs and include an adaptive guidance scheme.

**Outlook**   While we focused on sparse spatial observations, our framework could naturally extend to temporal observations in time-dependent PDEs, enabling spatiotemporal evolution from limited measurements. Second, function-space diffusion models could serve as a unifying methodology for diverse physical systems–enabling foundation models that can be trained once on multiple PDE and domain types and then adapted to various downstream tasks. Lastly, while FunDPS demonstrates promise for general PDE-based inverse problems, its performance on specialized inverse tasks such as MRI reconstruction [63] and full waveform inversion [64] remains to be seen.

## Acknowledgments

The authors want to thank Christopher Beckham for his efficient implementation of diffusion operators. Our thanks also extend to Jiahe Huang for her discussion on DiffusionPDE, and Yizhou Zhang for his assistance with baselines. Anima Anandkumar is supported in part by Bren endowed chair, ONR (MURI grant N00014-23-1-2654), and the AI2050 senior fellow program at Schmidt Sciences.

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

# A Diffusion Models in Function Spaces

## A.1 Forward Diffusion with Gaussian Random Fields

*Infinite-dimensional data* (such as functions defined on a continuous domain) require a careful definition of the diffusion process on them. We adopt a measure-theoretic approach following recent score-based generative models in function spaces [14]. Let $H$ be a separable Hilbert space of functions (the function space of interest), and let $\mu$ be the data distribution on $H$ (a probability measure over $H$ for our training data). In infinite dimensions, there is no Lebesgue density, so we work relative to a reference Gaussian measure. We introduce a centered Gaussian *prior* measure $\mathcal{N}(0, \mathbf{C})$ on $H$, with covariance operator $\mathbf{C}$ chosen to be self-adjoint, non-negative, and trace-class (so that $\mathcal{N}(0, \mathbf{C})$ is a well-defined GRF measure). We refer to $\mathcal{N}(0, \mathbf{C})$ as the GRF prior measure.

**Noising process**   Given a sample $a \sim \mu$, we perturb it by adding an independent Gaussian random function (drawn from the GRF prior) with appropriate variance. In other words, for a noise scale $\sigma \geq 0$, we define the noisy function at level $\sigma$ as

$$a_\sigma = a + \eta, \tag{15}$$

where $\eta \sim \mathcal{N}(0, \sigma^2 \mathbf{C})$ is a Gaussian random element in $H$ with covariance $\sigma^2 \mathbf{C}$. As $\sigma$ increases, more Gaussian noise is added to the function. For $\sigma = 0$, we have $a_0 = a$ (no noise), and at large $\sigma$, $a_\sigma$ is dominated by noise (In fact as $\sigma \to \infty$, $a_\sigma$ approaches a draw from the zero-mean GRF prior). This construction defines the forward diffusion process in function space in a *distributional* sense: it transforms the data distribution $\mu$ into a family of perturbed distributions $\mu_\sigma$, where $\mu_\sigma$ is the law of $a_\sigma = a + \eta$. Equivalently, $\mu_\sigma$ is the convolution of $\mu$ with the Gaussian measure $\mathcal{N}(0, \sigma^2 \mathbf{C})$. By varying $\sigma$ from 0 to some large value, we obtain a continuum (or a discrete set) of distributions bridging $\mu_0 = \mu$ and an almost pure noise distribution (when $\sigma$ is high). This is analogous to the forward noising process in standard diffusion models, but defined on an infinite-dimensional function space via GRFs.

## A.2 Score Function and Denoising Objective

With the forward process defined, we now consider the *score function* on the function space. At a given noise level $\sigma$, the score is defined as the gradient of the log-density of the perturbed distribution $\mu_\sigma$. In our infinite-dimensional setting, this gradient is understood with respect to the Gaussian reference measure (the GRF prior). Formally, for each $\sigma$, we define the score function $s(a, \sigma)$ as the $H$-valued gradient of $\log p_\sigma(a)$, where $p_\sigma$ is the density of $\mu_\sigma$ relative to $\mathcal{N}(0, \sigma^2 \mathbf{C})$. Intuitively, $s(a, \sigma)$ points in the direction in $H$ that *increases* the likelihood of $a$ under $\mu_\sigma$ the most. In finite dimensions, this recovers $\nabla_a \log p_\sigma(a)$; here we assume $s(a, \sigma)$ exists as an element of $H$ [14].

**Denoising score matching**   In practice, $s(a, \sigma)$ is unknown because the true data distribution $\mu$ (and hence $\mu_\sigma$) is unknown. Instead of trying to directly estimate the score, we train a *denoising model* $D_\theta(a_\sigma, \sigma)$ to recover the underlying clean function $a$ from a noisy sample $a_\sigma$. This is called *denoising score matching*, which is particularly convenient in function spaces. Following the EDM [16] training strategy, we sample pairs of clean and noisy functions and train $D_\theta$ to predict the clean input. In particular, given a sample $a \sim \mu$ and its noisy version $y = a + \eta$ with $\eta \sim \mathcal{N}(0, \sigma^2 \mathbf{C})$, we train $D_\theta$ to output $a$ (the ground truth) when given $(y, \sigma)$ as input. The training objective, averaged over the data distribution and noise, is formulated as a weighted mean squared error:

$$L(\theta) = \mathbb{E}_{a \sim \mu, \, \eta \sim \mathcal{N}(0, \sigma^2 \mathbf{C})} \left[ \lambda(\sigma) \, \| D_\theta(a + \eta, \, \sigma) - a \|_H^2 \right], \tag{16}$$

where $\lambda(\sigma)$ is a positive weighting function that balances the loss contributions across different noise levels. Lim et al. [14] shows that minimizing this *denoising objective* for all $\sigma$ is equivalent to learning the true score function on $H$. In fact, there is an explicit relationship between the optimal denoiser and the score operator in function space, analogous to Tweedie's formula. Given a noisy observation $y = a + \eta$ at scale $\sigma$, the score can be written as:

$$s(y, \sigma) = \frac{D_\theta(y, \sigma) - y}{\sigma^2}. \tag{17}$$

Thus, by training the model to minimize $L(\theta)$ across many noise levels, we are effectively teaching it to approximate the score operator in $H$. We implement the above training with a *discrete noise level*

*schedule* and random sampling of noise intensities, similar to the EDM methodology (Algorithm 1). We randomly sample $\sigma$ noise levels log-uniformly over $[\sigma_{\min}, \sigma_{\max}]$ as in Karras et al. [16]. As a result, $D_\theta$ becomes a function-space *denoiser* that can gradually refine a noisy input at any noise scale within the range.

---

**Algorithm 1** FunDPS Training (Training an unconditional diffusion model in function spaces)

---

**Require:** Data distribution $\mu$, GRF prior covariance $C$, noise-level distribution $p(\sigma)$
1: Initialize model parameters $\theta$
2: **repeat**
3:     $a \sim \mu,\ \sigma \sim p(\sigma)$                                   {Draw clean function and noise level}
4:     $\eta \sim \mathcal{N}(0, \sigma^2 \mathbf{C})$                                           {Sample GRF noise}
5:     $y \leftarrow a + \eta$                                             {Construct noisy sample}
6:     $\hat{a} \leftarrow D_\theta(y, \sigma)$                                     {Compute denoised prediction}
7:     $L \leftarrow \lambda(\sigma)\|\hat{a} - a\|_H^2$                                 {Compute training loss}
8:     Update parameters $\theta$ by minimizing $L$
9: **until** *converged*
10: **return** $D_\theta$

---

## A.3 Reverse Diffusion and Sampling

Once the denoising model (score model) is trained, we can generate new function samples from the learned distribution by running the diffusion process in reverse – starting from noise and iteratively *removing* noise. The key idea is to start with an initial random field drawn from the prior and then repeatedly apply the denoiser $D_\theta$ while decreasing $\sigma$ in stages. Here we adopt a discrete reverse diffusion approach aligned with EDM's deterministic solver. First, we choose a high noise level $\sigma_{\max}$ (e.g. the upper bound used in training) and sample an initial function $a_N \sim \mathcal{N}(0, ; \sigma_{\max}^2, \mathbf{C})$, i.e. a pure noise sample in $H$ drawn from the GRF prior. Then we define decreasing sequence of noise levels $\sigma_{\max} = \sigma_N > \sigma_{N-1} > \cdots > \sigma_1 > \sigma_0 \approx 0$, where the levels are spaced polynomially so that adjacent levels have small differences in terms of signal-to-noise ratio as in EDM [16]. The noise scheduler spans from the highest noise to zero ($\sigma_{\max}$ corresponds to the prior and $\sigma_0 = 0$ corresponds to a clean sample). At each step $i = N, N-1, \ldots, 1$, given the current noisy sample $a_i$ at noise level $\sigma_i$, we apply the denoiser to get $\hat{a} = D_\theta(a_i, \sigma_i)$, the model's estimate of the clean underlying function. It is followed by adding the $\sigma_{i-1}$ noise level to reach the next step's sample $a_{i-1}$ along with higher-order updates. After iterating down to the final level $\sigma_0 \to 0$, we obtain $a_0$, which is an approximate sample from the original data distribution $\mu$ (since no noise remains). We can then use $a_0$ as a newly generated function sample drawn from the learned generative model. We implement our framework using a deterministic sampler based on Euler's 2nd order method. Algorithm 2 without the FunDPS guidance corresponds to this unconditional sampling procedure.

## B Pseudocode for FunDPS

---

**Algorithm 2** FunDPS Sampler

---

**Require:** Observation $u$, forward operator $A$, denoising diffusion operator $D_\theta$, variance schedule $\{\sigma(t_i)\}_{i=0}^N$ with $\sigma(t_0) = 0$, guidance weights $\zeta$.
1: $a_N \sim \mathcal{N}(0, \mathbf{C})$ {*Initialize $a$ from GRF*}
2: **for** $i = N$ to 1 **do**
3:     $\hat{a}_0 \leftarrow D_\theta(a_i, \sigma(t_i))$                               {*Estimate $a_0$ by Tweedie's formula*}
4:     $d_i \leftarrow (a_i - \hat{a}_0)/\sigma(t_i)$                               {*Evaluate $\mathrm{d}a/\mathrm{d}t$ at $t_i$*}
5:     $a_{i-1} \leftarrow a_i + (\sigma(t_{i-1}) - \sigma(t_i))d_i$       {*Take Euler step from $\sigma(t_i)$ to $\sigma(t_{i-1})$*}
6:     **if** $\sigma(t_{i-1}) \neq 0$ **then**
7:         $\hat{a}_0' \leftarrow D_\theta(a_{i-1}, \sigma(t_{i-1}))$
8:         $d_i' \leftarrow (a_{i-1} - \hat{a}_0')/\sigma(t_{i-1})$
9:         $a_{i-1} \leftarrow a_i + (\sigma(t_{i-1}) - \sigma(t_i))(\frac{1}{2}d_i + \frac{1}{2}d_i')$      {*Apply 2nd-order correction*}
10:    **end if**
11:    $a_{i-1} \leftarrow a_{i-1} - \zeta \cdot \nabla_{a_i}\|u - A(\hat{a}_0')\|_{\mathcal{U}}^2$     {*Invoke the FunDPS guidance in Eq. (14)*}
12: **end for**
13: **return** $a_0$

---

# C  Measure-Theoretic Tweedie's Formula

In this section, we prove a generalization of Tweedie's formula which holds for separable Banach spaces. This will furnish us with a link between score functions [14] and conditional expectations, which is a key step in enabling function-space inverse problem solvers. We begin with some preliminaries on Gaussian measures before proceeding with our proof.

**Notation**   Throughout this section, $B$ will represent a separable Banach space. For two measures $\mu, \nu$ on $B$, we write $\mu \ll \nu$ if $\mu$ is absolutely continuous with respect to $\nu$, i.e., if $E \in \mathcal{B}(B)$ is Borel measurable and $\nu(E) = 0$, then $\mu(E) = 0$. We write $\mu \sim \nu$ are equivalent if $\nu \ll \nu$ and $\nu \ll \mu$. For any $h \in B$, we will write $T_h : B \to B$ for the translation map $x \mapsto x + h$. For an arbitrary Borel measure $\mu$ on $B$ we will write $\mu^h = (T_h)_\# \mu = \mu(\cdot - h)$ for the translated measure.

## C.1   Banach Space Gaussian Measures

We briefly review the key definitions necessary for our constructions. We refer to [50] for an in-depth treatment of this material. A centered Gaussian measure $\gamma$ on $B$ is a Borel probability measure such that the pushforward of $\gamma$ along any bounded linear functional $f \in B^*$ is a mean-zero Gaussian distribution on $\mathbb{R}$. Note that, as $B$ is separable, Fernique's theorem guarantees that we have $B^* \subset L^2(\gamma)$. The reproducing kernel Hilbert space (RKHS) $B^*_\gamma$ associated with $\gamma$ is the closure of $B^*$ with respect to the $L^2(\gamma)$ norm, i.e.,

$$B^*_\gamma = \overline{\{f \in X^*\}^{L^2(\gamma)}} \tag{18}$$

and for $f, g \in B^*_\gamma$ their inner product is $\langle f, g \rangle_{B^*_\gamma} = \int f(x)g(x)\, \mathrm{d}\gamma(x)$.

The measure $\gamma$ is uniquely determined by its covariance operator $C_\gamma : B^*_\gamma \to B^{**}$, defined by

$$C_\gamma(f)(g) = \int f(x)g(x)\, \mathrm{d}\gamma(x) \qquad \forall f \in B^*_\gamma, g \in B^*. \tag{19}$$

The Cameron-Martin space $H(\gamma) \subset X$ is defined as

$$H(\gamma) = C_\gamma(B^*_\gamma) = \{C_\gamma(f) \mid f \in B^*_\gamma\}. \tag{20}$$

Although $H(\gamma) \subset B^{**}$ in general, because $B$ is a separable Banach space we may take $H(\gamma) \subset B$. That is, we write $h = C_\gamma(f)$ for some $h \in B$ and $f \in B^*_\gamma$ if $f(g) = C_\gamma(f)(g)$ for all $g \in B^*$. Since every $h, k \in H(\gamma)$ are of the form $h = C_\gamma(\hat{h}), k = C_\gamma(\hat{k})$ for some $\hat{h}, \hat{k} \in B^*_\gamma$, the space $H(\gamma)$ has an induced inner product

$$\langle h, k \rangle_{H(\gamma)} = \langle C_\gamma(\hat{h}), C_\gamma(\hat{k}) \rangle_{H(\gamma)} = \langle \hat{h}, \hat{k} \rangle_{L^2(\gamma)}. \tag{21}$$

The space $H(\gamma)$ is a Hilbert space under this inner product. We will write $|h|_{H(\gamma)}$ for the associated norm. Note that we may equivalently define

$$|h|_{H(\gamma)} = \sup\{f(h) \mid f \in B^*, C_\gamma(f)(f) \leq 1\} \tag{22}$$

where the Cameron-Martin space $H(\gamma)$ may be identified as [50, Theorem 3.2.3]

$$H(\gamma) = \{h \in X \mid |h|_{H(\gamma)} < \infty\}. \tag{23}$$

Note that the map $C_\gamma : B^*_\gamma \to H(\gamma)$ is an isometric isomorphism. Since $H(\gamma)$ is a Hilbert space, the Riesz representation theorem furnishes us with a canonical isometry $R : H(\gamma)^* \to H(\gamma)$. Thus $H(\gamma)^* \simeq B^*_\gamma$. Somewhat more explicitly, we obtain an isometric isomorphism

$$J : B^*_\gamma \to H^* \qquad f \mapsto \langle C_\gamma(f), \cdot \rangle_{H(\gamma)}. \tag{24}$$

Note further that $(R \circ J)(f) = C_\gamma(f)$ for $f \in B^*_\gamma$ gives us an isometry $R \circ J : B^*_\gamma \to H(\gamma)$.

The following shows that the Cameron-Martin space $H(\gamma)$ is precisely those directions under which we may translate the measure $\gamma$ while remaining absolutely continuous [50, Corollary 2.4.3, Theorem 3.2.3]. Moreover, the celebrated Cameron-Martin formula allows us to explicitly calculate the associated density.

**Theorem C.1** (Cameron-Martin). *On a separable Banach space $B$, an element $h \in H(\gamma)$ is an element of the Cameron-Martin space if and only if $\gamma^h \sim \gamma$ are equivalent in the sense of being mutually absolutely continuous. In this case,*

$$\frac{\mathrm{d}\gamma^h}{\mathrm{d}\gamma}(x) = \exp\left(C_\gamma^{-1}(h)(x) - \tfrac{1}{2}|h|_{H(\gamma)}^2\right).$$

## C.2 Tweedie's Formula

We now proceed to give a proof of a generalized Tweedie's formula for centered Gaussian measures on Banach spaces. Let $X \sim \mu$ be a random variable on $B$ with distribution $\mu$ and let $Z \sim \gamma$ be distributed according to an independent centered Gaussian measure with covariance $C_\gamma$. Define a new random variable

$$Y = X + Z \tag{25}$$

whose distribution $\nu = \mu \star \gamma$ is obtained by the convolution of these two measures, i.e.,

$$\nu(E) = \int_B \gamma(E - x) \, \mathrm{d}\mu(x) \qquad \forall E \in \mathcal{B}(B). \tag{26}$$

Observe that conditioned on a fixed value of $X = x$, we have $Y \mid X = x \sim \gamma^x$ is distributed according to a Gaussian measure $\gamma^x$ with mean $x$ and covariance $C_\gamma$.

We begin by showing that $\nu \sim \gamma$ are equivalent when $\mu$ is supported on the CM space $H(\gamma)$. This is a generalization of [14, Theorem 1], who show an analogous claim for separable Hilbert spaces. The proof is essentially the same in both cases, but we include it here for the sake of completeness.

**Proposition C.2.** *Suppose $\mu(H(\gamma)) = 1$. Then, $\nu \sim \gamma$ are equivalent.*

**Proof.** Let $E \in \mathcal{B}(B)$ be an arbitrary Borel set. Suppose that $\gamma(E) = 0$. By Theorem C.1, $\gamma^x \sim \gamma$ for $\mu$-almost every $x$, and hence $\gamma^x(E) = 0$ for $\mu$-almost every $x$. It follows that

$$\nu(E) = \int_B \gamma^x(E) \, \mathrm{d}\mu(x) = 0.$$

Conversely, suppose $\nu(E) = 0$. It follows that $\gamma^x(E) = 0$ for $\mu$-almost every $x$. Since $\mu(H(\gamma)) = 1$, Theorem C.1 shows the measures $\gamma^x$ and $\gamma$ are almost surely equivalent and thus $\gamma(E) = 0$. $\square$

Hence, if $\mu(H(\gamma)) = 1$, the Radon-Nikodym theorem provides us with a Borel measurable $\phi : B \to \mathbb{R}$ with

$$\frac{\mathrm{d}\nu}{\mathrm{d}\gamma}(y) = \exp(\phi(y)) \qquad \gamma\text{-a.e. } y \in B.$$

We henceforth assume this is the case. Assume further that $\phi$ is Fréchet differentiable along $H(\gamma)$. The score of $\nu$ is defined as

$$D_{H(\gamma)}\phi : B \to H(\gamma)^*.$$

That is, $D_{H(\gamma)}\phi = D_{H(\gamma)} \log \frac{\mathrm{d}\nu}{\mathrm{d}\gamma}$ is the logarithmic derivative of the density of $\nu$ along $H(\gamma)$. The value $[D_{H(\gamma)}\phi](x)(h)$ is the derivative of $\phi$ at $x$ in the direction $h \in H(\gamma)$. We refer to [14] for a further discussion of this notion of a score and the differentiability assumption.

In the following lemma, we prove that the density of $\nu$ with respect to the noise measure $\gamma$ can also be understood in terms of the corresponding conditional distributions $\gamma^x$. This lemma will be used to aid our later calculations.

**Lemma C.3.** *Assume that $\mu(H(\gamma)) = 1$. For $\gamma$-almost every $y \in B$, we have*

$$\frac{\mathrm{d}\nu}{\mathrm{d}\gamma}(y) = \int_B \frac{\mathrm{d}\gamma^x}{\mathrm{d}\gamma}(y) \, \mathrm{d}\mu(x) = \mathbb{E}_{x \sim \mu}\left[\frac{\mathrm{d}\gamma^x}{\mathrm{d}\gamma}(y)\right]. \tag{27}$$

*Moreover, for $\gamma$-almost every $y \in B$,*

$$\frac{\mathrm{d}\gamma}{\mathrm{d}\nu}(y) = \left(\frac{\mathrm{d}\nu}{\mathrm{d}\gamma}(y)\right)^{-1}. \tag{28}$$

**Proof.** Fix a measurable $E \in \mathcal{B}(B)$. Since $\mu(H(\gamma)) = 1$, then $\mu$-almost surely we have that $\gamma^x \ll \gamma$. Hence, by the Radon-Nikodym theorem, the density $(\,\mathrm{d}\gamma^x / \,\mathrm{d}\gamma)(y)$ exists $\mu$-almost everywhere.

Now, recall that $\nu = \mu \star \gamma$ is a convolution of measures, so that

$$\nu(E) = \int_B \gamma(E - x) \,\mathrm{d}\mu(x) \tag{29}$$

$$= \int_B \gamma^x(E) \,\mathrm{d}\mu(x) \tag{30}$$

$$= \int_B \int_E \frac{\mathrm{d}\gamma^x}{\mathrm{d}\gamma}(y) \,\mathrm{d}\gamma(y) \,\mathrm{d}\mu(x) \tag{31}$$

$$= \int_E \int_B \frac{\mathrm{d}\gamma^x}{\mathrm{d}\gamma}(y) \,\mathrm{d}\mu(x) \,\mathrm{d}\gamma(y). \tag{32}$$

where the last equality follows by Tonelli's theorem and the fact that the densities are nonnegative. By Proposition C.2, $\nu \ll \gamma$ and so by the Radon-Nikodym theorem the density $\mathrm{d}\nu/\,\mathrm{d}\gamma$ is uniquely defined up to a set of $\gamma$-measure zero. Thus,

$$\frac{\mathrm{d}\nu}{\mathrm{d}\gamma}(y) = \int_B \frac{\mathrm{d}\gamma^x}{\mathrm{d}\gamma}(y) \,\mathrm{d}\mu(x) \qquad \gamma\text{-a.e. } y \in B \tag{33}$$

as claimed. The second claim follows because $\nu \sim \gamma$ under the assumption $\mu(H(\gamma)) = 1$. $\qquad\square$

We now proceed to calculate $\mathbb{E}[X|Y = y]$. First, we directly calculate this using the definition of a conditional expectation. Note that this proof follows closely a calculation shown in [43], Appendix F.1.

**Proposition C.4.** *Suppose that $\mu(H(\gamma)) = 1$. Then, for $\nu$-almost every $y \in B$, the conditional expectation is given by*

$$\mathbb{E}[X \mid Y = y] = \frac{\mathrm{d}\gamma}{\mathrm{d}\nu}(y)\mathbb{E}_{x\sim\mu}\left[x\frac{\mathrm{d}\gamma^x}{\mathrm{d}\gamma}(y)\right]. \tag{34}$$

**Proof.** Write $f(y)$ for the right-hand side of (34) and let $A \in \sigma(Y)$ be a $Y$-measurable event. We show $\mathbb{E}_{y\sim\nu}[\mathbb{1}_A f(y)] = \mathbb{E}_{x\sim\mu}[\mathbb{1}_A x]$, from which the claim follows. Indeed,

$$\int_A f(y)d\nu(y) = \int_B \int_A f(y)d\gamma^{\tilde{x}}(y)d\mu(\tilde{x}) \tag{35}$$

$$= \int_B \int_B \int_A x\frac{\mathrm{d}\gamma}{\mathrm{d}\nu}(y)\frac{\mathrm{d}\gamma^x}{\mathrm{d}\gamma}(y) \,\mathrm{d}\gamma^{\tilde{x}}(y) \,\mathrm{d}\mu(\tilde{x}) \,\mathrm{d}\mu(x) \tag{36}$$

$$= \int_B \int_B \int_A x\frac{\mathrm{d}\gamma}{\mathrm{d}\nu}(y)\frac{\mathrm{d}\gamma^x}{\mathrm{d}\gamma}(y)\frac{\mathrm{d}\gamma^{\tilde{x}}}{\mathrm{d}\gamma}(y) \,\mathrm{d}\gamma(y) \,\mathrm{d}\mu(\tilde{x}) \,\mathrm{d}\mu(x) \tag{37}$$

$$= \int_B \int_A x\frac{\mathrm{d}\gamma}{\mathrm{d}\nu}(y)\frac{\mathrm{d}\gamma^x}{\mathrm{d}\gamma}(y)\left[\int_B \frac{\mathrm{d}\gamma^{\tilde{x}}}{\mathrm{d}\gamma}(y) \,\mathrm{d}\mu(\tilde{x})\right] \,\mathrm{d}\gamma(y) \,\mathrm{d}\mu(x) \tag{38}$$

$$= \int_B \int_A x\frac{\mathrm{d}\gamma}{\mathrm{d}\nu}(y)\frac{\mathrm{d}\gamma^x}{\mathrm{d}\gamma}(y)\frac{\mathrm{d}\nu}{\mathrm{d}\gamma}(y) \,\mathrm{d}\gamma(y) \,\mathrm{d}\mu(x) \tag{39}$$

$$= \int_A x\left[\int_B \frac{\mathrm{d}\gamma^x}{\mathrm{d}\gamma}(y) \,\mathrm{d}\gamma(y)\right] \,\mathrm{d}\mu(x) \tag{40}$$

$$= \mathbb{E}_{x\sim\mu}[\mathbb{1}_A x]. \tag{41}$$

which completes the proof. $\qquad\square$

We now proceed to calculate the Cameron-Martin space gradient of the score. In particular, we show that it is equal to the same expression we obtained in Proposition C.4. This requires an assumption on the measure $\mu$ to ensure that the derivatives of $\mathrm{d}\gamma^x / \,\mathrm{d}\gamma$ are bounded by an integrable function in order to justify a derivative-integral exchange. Using the continuity of $C_\gamma$, the condition in Eq. (43) can be relaxed to finding an integrable $\psi \in L^1(\gamma)$ such that for all $y \in B$ and $\mu$-almost every $x$,

$$|x|_{H(\gamma)} \exp\left(|C_\gamma^{-1}|_{B_\gamma^*}|x|_{H(\gamma)}|y|_B - \tfrac{1}{2}|x|_{H(\gamma)}^2\right) \leq \psi(x). \tag{42}$$

While this condition depends on the specific choice of $\mu$, it will be satisfied when e.g. $\mu$ is compactly supported in $H(\gamma)$ or when $\mu$ has tails which decay sufficiently fast.

**Proposition C.5.** *Assume $\mu(H(\gamma)) = 1$ and that $\log \frac{d\nu}{d\gamma}$ is Fréchet differentiable along $H(\gamma)$. Consider the score $D_{H(\gamma)} \log \frac{d\nu}{d\gamma} : B \to H(\gamma)^*$. Let $R : H(\gamma)^* \to H(\gamma)$ be the Riesz isometry. Assume further that there exists a non-negative function $\psi \in L^1(\mu)$ such that for all $y \in B$ and $\mu$-almost every $x \in B$,*

$$\left| D_{H(\gamma)} \frac{d\gamma^x}{d\gamma}(y) \right|_{H(\gamma)^*} \leq \psi(x). \tag{43}$$

*Then, the Cameron-Martin score*

$$R\left( D_{H(\gamma)} \log \frac{d\nu}{d\gamma} \right) : B \to H(\gamma) \tag{44}$$

*is given for $\nu$-almost every $y \in B$ as*

$$R\left( D_{H(\gamma)} \log \frac{d\nu}{d\gamma}(y) \right) = \frac{d\gamma}{d\nu}(y) \mathbb{E}_{x \sim \mu} \left[ x \frac{d\gamma^x}{d\gamma}(y) \right]. \tag{45}$$

**Proof.** Since we assume the score of $\nu$ is Fréchet differentiable, we may apply the chain rule to see that for $y \in B$,

$$D_{H(\gamma)} \log \frac{d\nu}{d\gamma}(y) = \frac{d\gamma}{d\nu}(y) D_{H(\gamma)} \frac{d\nu}{d\gamma}(y). \tag{46}$$

Moreover, by the Cameron-Martin formula, if $x \in H(\gamma)$, then

$$\left[ D_{H(\gamma)} \log \frac{d\gamma^x}{d\gamma} \right](y) = J(C_\gamma^{-1}(x)) \tag{47}$$

where $J : B_\gamma^* \to H^*$ is the isomorphism defined in Eq. (24). Note this isomorphism is required as $C_\gamma^{-1}(x) \in B^*\gamma$ is an element of the RKHS $B_\gamma^*$, whereas the Fréchet derivative is an element of $H(\gamma)^*$. Using Lemma C.3 and the assumption that there exists a dominating function $\psi \in L^1(\mu)$, we may use the Leibniz integral rule to calculate that

$$\left[ D_{H(\gamma)} \frac{d\nu}{d\gamma} \right](y) = \int_B J(C_\gamma^{-1}(x)) \frac{d\gamma^x}{d\gamma}(y) \, d\mu(x) \tag{48}$$

in the sense that

$$\left[ D_{H(\gamma)} \frac{d\nu}{d\gamma} \right](y)(h) = \int_B J\left( C_\gamma^{-1}(x) \right)(h) \frac{d\gamma^x}{d\gamma}(y) \, d\gamma(x) \qquad \forall h \in H(\gamma). \tag{49}$$

Using the fact that $R$ is bounded and $R \circ J = C_\gamma$, we obtain

$$R\left( D_{H(\gamma)} \frac{d\nu}{d\gamma}(y) \right) = \int_B x \frac{d\gamma^x}{d\gamma}(y) \, d\gamma(x) \tag{50}$$

$$= \mathbb{E}_{x \sim \gamma} \left[ x \frac{d\gamma^x}{d\gamma}(y) \right]. \tag{51}$$

Combined with Eq. (46), this yields the claim. $\qquad \square$

Combining Proposition C.4 and Proposition C.5 yields the Banach space generalization of Tweedie's formula, which concludes the proof for Theorem 3.1.

### C.3  Special Case: Euclidean Setting

Here, we will informally replicate our proof of Tweedie's formula in the special case of $B = \mathbb{R}^n$ to provide some intuition and to sanity check this result. In this case we suppose everything admits densities, so that $\nu = p_Y(y), \mu = p_X(x)$, and $\gamma = p_Z(z) = \mathcal{N}(z \mid 0, C)$. Moreover $p_{Y|X}(y \mid x) = \mathcal{N}(y \mid x, C)$. While many of these steps in this section can be done in a more straightforward fashion when $B = \mathbb{R}^n$, we purposefully follow the structure of our previous calculations.

In this setting, we seek to compute

$$\nabla_y \log \frac{\mathrm{d}\nu}{\mathrm{d}\gamma}(y) = \nabla_y \log \frac{p_Y(y)}{p_Z(y)}. \tag{52}$$

as the score is now taken with respect to the noise measure $p_Z(y)$. Now, using the densities, we may calculate in an analogous fashion

$$\nabla_y \log \frac{p_Y(y)}{p_Z(y)} = \frac{p_Z(y)}{p_Y(y)} \nabla_y \left( \frac{1}{p_Z(y)} \int_{\mathbb{R}^n} p_Y(y \mid x) p_X(x) \, \mathrm{d}x \right) \tag{53}$$

$$= \frac{p_Z(y)}{p_Y(y)} \int_{\mathbb{R}^n} \nabla_y \log \left( \frac{p_Y(y \mid x)}{p_Z(y)} \right) \frac{p_Y(y \mid x)}{p_Z(y)} p(x) \, \mathrm{d}x \tag{54}$$

$$= \frac{p_Z(y)}{p_Y(y)} \int_{\mathbb{R}^n} \left( C^{-1}(x - y) + C^{-1}(y) \right) \frac{p_Y(y \mid x)}{p_Z(y)} p(x) \, \mathrm{d}x \tag{55}$$

$$= \frac{p_Z(y)}{p_Y(y)} \int_{\mathbb{R}^n} C^{-1}(x) \frac{p_Y(y \mid x)}{p_Z(y)} p(x) \, \mathrm{d}x. \tag{56}$$

This expression is the finite-dimensional analogue of the one we obtain in Eq. (48). This yields

$$C \nabla_y \log \frac{p_Y(y)}{p_Z(y)} = \frac{p_Z(y)}{p_Y(y)} \int_{\mathbb{R}^n} x \frac{p_Y(y \mid x)}{p_Z(y)} p(x) \, \mathrm{d}x. \tag{57}$$

On the other hand, we may explicitly calculate the conditional expectation by

$$\mathbb{E}[X \mid Y = y] = \int_{\mathbb{R}^n} x p_{X \mid Y}(x \mid y) \, \mathrm{d}x \tag{58}$$

$$= \frac{\int_{\mathbb{R}^n} x p_{Y \mid X}(y \mid x) p_X(x) \, \mathrm{d}x}{p_Y(y)} \tag{59}$$

$$= \frac{p_Z(y)}{p_Y(y)} \int_{\mathbb{R}^n} x \frac{p_{Y \mid X}(y \mid x)}{p_Z(y)} p_X(x) \, \mathrm{d}x \tag{60}$$

$$= C \nabla_y \log \frac{p_Y(y)}{p_Z(y)}, \tag{61}$$

Let us take a step further towards the standard expression of Tweedie's formula. The gradient $\nabla_y \log \frac{p_Y(y)}{p_Z(y)}$ can be expanded as $\nabla_y \log p_Y(y) - \nabla_y \log p_Z(y)$. For $p_Z(y) = \mathcal{N}(z \mid 0, C)$, we have $\nabla_y \log p_Z(y) = -C^{-1}y$. Substituting, this becomes the more familiar expression

$$\mathbb{E}[X \mid Y = y] = y + C \nabla_y \log p_Y(y). \tag{62}$$

## D   Inverse PDE Solver with an FDM Forward Operator

In order to show the diverse adaptability of our methodology, we further apply our framework to solve inverse PDE problems by using a Finite Difference Method (FDM)-based forward operator. In this problem, we only model the prior distribution on the coefficient space using the function-space diffusion model. We then reconstruct the initial states from full, noisy, and sparse observations of the solution space. Specifically, the forward operator here is

$$\boldsymbol{A}(\boldsymbol{a}) = \texttt{FDM\_Solve}(-\nabla[\boldsymbol{a}\nabla\boldsymbol{u}] = 1), \tag{63}$$

and we formulate the inverse problem as estimating initial state $\boldsymbol{a}$ from corrupted observations of solution state $\boldsymbol{u}$. Our goal is to sample from $p(\boldsymbol{a}|\boldsymbol{u})$ by applying the function-space reverse diffusion steps with a guidance term $\nabla_{\boldsymbol{a}}\|\boldsymbol{A}(\boldsymbol{a}) - \boldsymbol{u}\|$. Since $\boldsymbol{A}$ is a standard FDM solver, we can compute gradients via automatic differentiation. We present qualitative examples in Figure 5 and show in Table 2 that our method can achieve small relative $L^2$-errors in these challenging cases with both FDM as forward operator and joint learning. In practice, considering similar or better performance, we use a joint learning approach for all our experiments as the inference speed is considerably faster, since the FDM-based method requires backpropagating through the linear solve `FDM_Solve`.

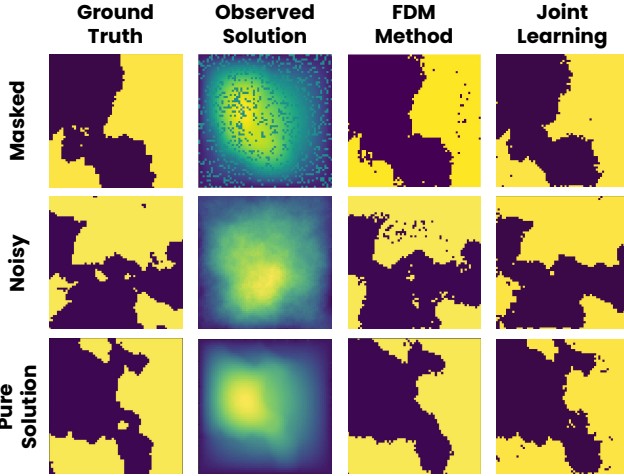

Figure 5: Reconstruction of coefficient functions from partially-observed Darcy Flow problems.

Table 2: The reconstruction error on the inverse Darcy Flow problem with diverse settings.

|  | Steps ($N$) | Time | Corruption | | |
|---|---|---|---|---|---|
|  |  |  | None | Noisy | Masked |
| FDM | 50 | 57s | 4.28% | 4.67% | 4.47% |
| Joint Learning | 500 | 15s | 3.64% | 6.32% | 3.95% |

## E   Detailed Dataset Description

**Darcy Flow**   Darcy Flow is a fundamental model that describes the flow of a viscous incompressible fluid through a porous medium. The governing equations are given by:

$$-\nabla \cdot (a(x)\nabla u(x)) = f(x), \qquad x \in (0,1)^2, \tag{64}$$

with constant forcing $f(x) = 1$ and zero boundary conditions. We follow the strategies in Li et al. [21] to generate coefficient functions $a \sim h_{\#}\mathcal{N}(0, (-\Delta + 9\mathbf{I})^{-2})$, where $h\colon \mathbb{R} \to \mathbb{R}$ is set to be 12 for positive numbers and 3 otherwise.

**Poisson Equation**    The Poisson equation describes steady-state diffusion processes:

$$\nabla^2 u(x) = a(x), \quad x \in (0,1)^2, \tag{65}$$

with homogeneous Dirichlet boundary conditions $u|_{\partial\Omega} = 0$. We generate coefficient fields $a(x)$ from Gaussian random fields $\mathcal{N}(0, (-\Delta + 9\mathbf{I})^{-2})$. The PDE guidance function is $f = \nabla^2 u - a$.

**Helmholtz Equation**    The Helmholtz equation models wave propagation in heterogeneous media:

$$\nabla^2 u(x) + k^2 u(x) = a(x), \quad x \in (0,1)^2, \tag{66}$$

with $k = 1$ and Dirichlet boundary conditions $u|_{\partial\Omega} = 0$. Coefficient fields $a(x)$ are GRFs generated as in [17]. The PDE guidance function is $f = \nabla^2 u + k^2 u - a$.

**Navier-Stokes Equations**    We further evaluate the performance on the Navier-Stokes equations by generating its initial and terminal states as in [21]. In particular, we consider the evolution of a velocity field $\boldsymbol{u}(x,t)$ over time given by

$$\partial_t \boldsymbol{w}(x,t) + \boldsymbol{u}(x,t) \cdot \nabla \boldsymbol{w}(x,t) = \nu \Delta \boldsymbol{w}(x,t) + f(x), \quad x \in (0,1)^2, t \in (0,T], \tag{67}$$

$$\nabla \cdot \boldsymbol{u}(x,t) = 0, \quad x \in (0,1)^2, t \in [0,T], \tag{68}$$

$$\boldsymbol{u}(x,0) = \boldsymbol{a}(x), \quad x \in (0,1)^2, \tag{69}$$

where $\boldsymbol{w} = \nabla \times \boldsymbol{u}$ is the vorticity; $\nu = \frac{1}{1000}$, viscosity; and $f$, a fixed forcing term. The initial condition $\boldsymbol{a}(x)$ is sampled from $\mathcal{N}(0, 7^{3/2}(-\Delta + 49\mathbf{I})^{-5/2})$. The forcing term is defined as $f(x) = \frac{1}{10}\left(\sin(2\pi(x_1 + x_2)) + \cos(2\pi(x_1 + x_2))\right)$. We simulate the PDE for $T = 1$ using a pseudo-spectral method. It should be noted that the PDE guidance formulated in Huang et al. [17] is invalid:

$$\nabla \cdot \boldsymbol{w} = \nabla \cdot (\nabla \times \boldsymbol{u}) = 0 \tag{70}$$

Furthermore, due to the lack of information, calculating a PDE loss is non-trivial here. While the experiments in this work use the original incorrect formulation for consistency with prior benchmarks, we anticipate minimal impact on the final results given the relatively small loss weight $\lambda_{PDE}$.

**Navier-Stokes Equations with Boundary Conditions (BCs)**    We study bounded flow around cylindrical obstacles, governed by:

$$\partial_t \boldsymbol{v}(x,t) + \boldsymbol{v}(x,t) \cdot \nabla \boldsymbol{v}(x,t) = -\nabla p + \nu \nabla^2 \boldsymbol{v}(x,t), \quad x \in \Omega, t \in (0,T], \tag{71}$$

$$\nabla \cdot \boldsymbol{v}(x,t) = 0, \quad x \in \Omega, t \in (0,T], \tag{72}$$

with $\nu = 0.001$, $\rho = 1.0$, and no-slip boundaries on $\partial\Omega_{\text{left,right,cylinder}}$. The domain contains randomly placed cylinders. We learn the joint distribution of $v_0$ and $v_T$ at $T = 4$. Its original PDE guidance has the same error as the non-bounded case.

# F    Detailed Experiment Setup

**Datasets**    We validate our approach by solving both forward and inverse problems on five different PDE problems. These PDEs include Darcy Flow, Poisson, Helmholtz, and Navier-Stokes with and without boundary conditions. We follow the same strategy as in DiffusionPDE [17] to generate datasets, where we prepare $50,000$ training samples and $1,000$ test samples for each PDE. The Navier-Stokes equation with boundary conditions specifically consists of $14,000$ train and $1,000$ test samples. The resolution is $128 \times 128$, and in some settings we downsample the data by $2\times$. For quantitative comparisons, error rates are calculated using the $L^2$ relative error between the predicted and true solutions, except for the inverse Darcy Flow problem, where we use the binary error rate.

**Implementation**    We adopt a 4-level U-shaped neural operator architecture [52] as the denoiser $\boldsymbol{D}_\theta$, which has 54M parameters, similar to DiffusionPDE's network size. The network is trained using $50,000$ training samples for 200 epochs. For the multi-resolution training, we begin training on a coarser grid ($64 \times 64$) for 200 epochs, then switch to a higher resolution ($128 \times 128$) for 100 epochs. The hyperparameters we used for training and inference are listed in Table 3. We source the quantitative results of deterministic baselines from DiffusionPDE [17]'s table.

Table 3: Hyperparameters of Choice.

| Hyperparameter | Value |
|---|---|
| learning_rate | 0.0001 |
| learning_rate_warmup | 5 million samples |
| ema_half_life | 0.5 million samples |
| dropout | 0.13 |
| rbf_scale | 0.05 |
| sigma_max | 80 |
| sigma_min | 0.002 |
| rho | 7 |

**Speed comparison**  All the experiments are conducted using a single NVIDIA RTX 4090 GPU. To determine per-sample inference time, we averaged batch inference time over 10 runs and divided by the batch size. Batch sizes were optimized to fully utilize GPU memory; specifically, for $128 \times 128$ data, these were 13 for FunDPS and 8 for DiffusionPDE.

# G   Implementation Details of the Guidance Mechanism

For inference, we tuned the guidance strength $\zeta$ on a small validation set, resulting in the values shown in Table 4. We noted that PDE loss calculations are unreliable in early sampling stages due to high noise levels. Hence empirically, we only apply PDE loss when $\sigma_t < 1$. Furthermore, to ensure smooth convergence to the posterior, we found it beneficial to dial down the guidance weights as the noise level decreases. Therefore, we implement a simple but effective scheduling scheme for the guidance weights of both observation and PDE loss:

$$\tilde{\zeta}_t = \begin{cases} \sigma_t \zeta & \text{if } \sigma_t < 1 \\ \zeta & \text{if } \sigma_t \geq 1 \end{cases}$$

We use the Huber loss for PDE guidance instead of mean squared error because it provides robustness against potential outliers caused by finite difference approximation errors, which improves the stability of gradient updates. We investigated the method's sensitivity to the guidance strength in Appendix H.6.

Table 4: Guidance strength $\zeta$ used for each PDE problem.

| | Darcy Flow | | Poisson | | Helmholtz | | Navier-Stokes | | Navier-Stokes with BCs | |
|---|---|---|---|---|---|---|---|---|---|---|
| | Forward | Inverse | Forward | Inverse | Forward | Inverse | Forward | Inverse | Forward | Inverse |
| Observation Loss Type | MSE | MSE | MSE | MSE | MSE | L2 | MSE | L2 | MSE | L2 |
| Observation Loss Weight | 10000 | 50000 | 10000 | 20000 | 10000 | 5000 | 5000 | 7500 | 3000 | 2000 |
| PDE Loss Type | Huber | Huber | Huber | Huber | Huber | Huber | Huber | Huber | Huber | Huber |
| PDE Loss Weight | 0 | 0 | 0 | 0 | 1 | 1 | 1 | 10 | 100 | 15 |

# H  Additional Experiments

In this section, we will present additional experiments to demonstrate key aspects of our model. Due to time constraints, we primarily focus on two representative PDEs: Darcy Flow and Navier-Stokes equation. These systems are of significant interest and range from smooth elliptic problems to highly nonlinear dynamics.

## H.1  Plug-and-play inverse solvers

In order to show the adaptability of the framework, we further test our method with various guidance methods. Namely, Table 5 and Figure 6 demonstrate the reconstruction results on Poisson PDE equation with various inverse solvers [40, 39, 65] on different priors. FunDPS (Function Space + DPS) consistently outperforms other methods within a smaller number of steps.

The significant underperformance of DDNM and DAPS in our setting is primarily due to the extreme sparsity of observation data. DDNM relies on projecting the sample onto measurement subspace at each step. With only 3% of observation points, the measurement subspace is extremely low-dimensional compared to the overall function space. As a result, the projection provides very weak guidance and leads to poor reconstruction. The core issue of DAPS comes from the localness of guided updates during Langevin dynamics. Only the points with observation are updated and others are just added with noise. This makes the intermediate state after Langevin dynamics discontinuous and out-of-distribution, which hinders performance. If we increase the number of sampling steps to 20,000, the accuracy can match DPS, but with 100x more time.

Table 5: The relative errors for Poisson equation under varying priors and plug-and-play inverse solvers. FunDPS corresponds to the intersection of Function Space prior with DPS solver. FunDPS results are based on 500 steps, whereas other methods are performed with at least 2000 steps.

| Inverse Solvers | Function Space | | Euclidean Space | |
|---|---|---|---|---|
| | Forward | Inverse | Forward | Inverse |
| DPS [40] | **1.99%** | **20.47%** | 4.88% | 21.10% |
| DDNM [65] | 10.17% | 41.67% | 20.66% | 43.84% |
| DAPS [39] | 40.35% | 77.72% | 492.6% | 274.8% |

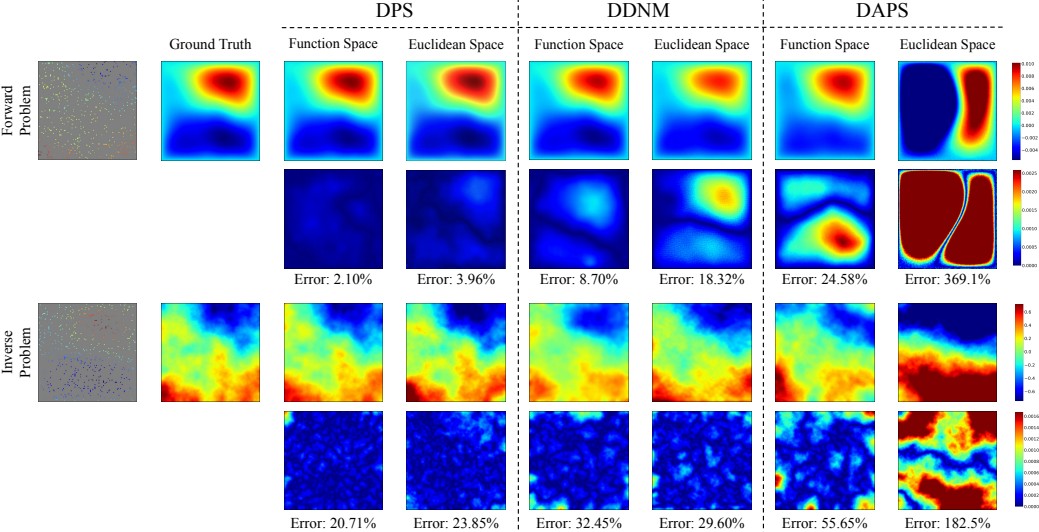

Figure 6: The qualitative results of Table 5 with Poisson equation. First and second rows correspond to the forward problem reconstructions and error maps, respectively. The third and fourth rows correspond to the inverse problem reconstructions and error maps, respectively. Relative errors are also reported for this specific data under each error map, where FunDPS achieves the minimum among all the tasks.

## H.2 Fully-observed problems

Deep learning approaches have been extensively researched for solving classical forward and inverse problems, where either the initial state or the final state is fully known. While the deterministic baselines are powerful in these tasks, our method still demonstrates superior performance in 3 out of 4 cases, highlighting the strong modeling capabilities of our framework. Ours also outperforms DiffusionPDE in all cases, shown in Table 6.

Table 6: Comparison of different methods on forward and inverse problems with full observation. Error rates are calculated using the $L^2$ relative error between the predicted and true solutions, except for the Darcy Flow inverse problem, where a binary error rate is used. FunDPS significantly outperforms the fixed-resolution diffusion baseline and achieves top performance even when compared to deterministic baselines in 3 out of 4 cases. The best results are highlighted in **bold**.

|  | Steps ($N$) | Darcy Flow | | Navier-Stokes | |
|---|---|---|---|---|---|
|  |  | Forward | Inverse | Forward | Inverse |
| FunDPS (ours) | 200 | 1.1% | 4.2% | 4.9% | 7.8% |
| FunDPS (ours) | 500 | 1.4% | 3.0% | 3.0% | 7.0% |
| FunDPS (ours) | 2000 | **0.9%** | **2.1%** | 1.6% | **6.6%** |
| DiffusionPDE[6] | 2000 | 2.9% | 13.0% | 2.4% | 8.4% |
| FNO | - | 5.3% | 5.6% | 2.3% | 6.8% |
| PINO | - | 4.0% | **2.1%** | **1.1%** | 6.8% |
| DeepONet | - | 12.3% | 8.4% | 25.6% | 19.6% |
| PINN | - | 15.4% | 10.1% | 27.3% | 27.8% |

## H.3 Multi-resolution training

Our multi-resolution training combines efficient low-resolution learning with high-resolution finetuning. We first train for 200 epochs on 64×64 resolution data to learn coarse features, then train for 100 epochs on 128×128 resolution to capture fine details. This approach leverages the resolution independence of neural operators while reducing computational costs.

Table 7 compares training configurations across resolutions. Direct high-resolution training performs well but requires substantially more parameters and computing resources. Our multi-resolution approach achieves comparable or better performance while maintaining the smaller model size, reducing total GPU hours by about 25%.

Table 7: Comparison of different training resolution strategies. "Mixed" refers to our two-phase approach, which achieves comparable or better performance to training directly at high resolution while using significantly fewer parameters. We use 500 steps for evaluation.

| Training Res. | Inference Res. | # Params | Darcy Flow | | Navier-Stokes | |
|---|---|---|---|---|---|---|
|  |  |  | Forward | Inverse | Forward | Inverse |
| 64 | 64 | 54M | 3.03% | 6.75% | 3.20% | 8.85% |
| 64 | 128 | 54M | 3.64% | 5.24% | 3.81% | 8.48% |
| 128 | 128 | 184M | 2.74% | 5.03% | 3.35% | 8.20% |
| Mixed | 128 | 54M | 2.49% | 5.18% | 3.32% | 8.16% |

## H.4 Multi-resolution inference

Our multi-resolution sampling pipeline is shown in Figure 3b. It includes two stages with noise increase in between. The first stage is a complete diffusion sampling at low resolution, from $\sigma_{\max}^{(1)}$ to $\sigma_{\min}^{(1)}$, over $t_{\mathrm{up}}$ steps. Samples are then upscaled (e.g., using bicubic interpolation, though our

---

[6]We again found reproducibility issues with DiffusionPDE. Our reproduced results are here. Please refer to Appendix I for details.

method is not sensitive to this choice). To counter upscaling artifacts and initiate a subsequent high-resolution diffusion, noise is added at $\sigma = \sigma_{\max}^{(2)}$. Subsequently, a second diffusion sampling process is performed. Empirically, we found that $\sigma_{\min}^{(2)}$ in the initial low-resolution stage need not be very low, as details are refined in the second stage. Furthermore, $\sigma_{\max}^{(2)}$ for initiating the high-resolution process is typically set to a modest value (e.g., in the 1–10 range), considerably lower than the initial $\sigma_{\max}^{(1)} = 80$, as we want to keep the existing information in the low-resolution sample. We provide the multi-resolution inference results in Figure 4. Table 8 further shows that FunDPS generalizes across resolutions, while DiffusionPDE fails to transfer and must be retrained for each fixed grid.

Table 8: Results on Darcy Flow under different training–inference resolution pairs. DiffPDE-2000 and DiffPDE-500 represent 2000 and 500 steps for Diffusion PDE, respectively. FunDPS, using 500 steps, achieves superior generalization and maintains accuracy under multi-resolution settings, whereas DiffusionPDE fails to generalize between grids.

| Training | Inference | DiffPDE-2000 | DiffPDE-500 | FunDPS |
|---|---|---|---|---|
| 64 | 64 | 6.38% | 7.96% | **3.03%** |
| 64 | 128 | 33.72% | 35.57% | **3.64%** |
| 128 | 128 | 6.07% | 4.60% | **2.74%** |
| Mixed | 128 | 3.83% | 4.53% | **2.49%** |

## H.5   Number of observations

We investigate how the number of observations affects model performance in both forward and inverse Darcy Flow problems. We use 500 steps for evaluation. Figure 7 shows that performance improves consistently as we increase the number of observed points from 100 (0.6%) to 2000 (12%) of the spatial domain. It is worth noting that our method can achieve reasonable accuracy even with extremely sparse observations, demonstrating its effectiveness.

We further analyze the performance of FunDPS and DiffusionPDE under different observation sparsities on both forward and inverse Darcy Flow problems. As shown in Table 9, FunDPS maintains high accuracy even with extremely sparse observations (as low as $0.5\%$ of spatial points), while DiffusionPDE performance degrades significantly under sparse settings.

Table 9: Comparison of FunDPS and DiffusionPDE across varying numbers of observations on Darcy Flow. DiffPDE-2000 and DiffPDE-500 represent 2000 and 500 steps for Diffusion PDE, respectively. FunDPS, using 500 steps, maintains <10% relative error even with as few as 0.5% observed points.

| # of Obs. | Forward | | | Inverse | | |
|---|---|---|---|---|---|---|
| | DiffPDE-2000 | DiffPDE-500 | FunDPS | DiffPDE-2000 | DiffPDE-500 | FunDPS |
| 100 | **7.2%** | 13.8% | **7.2%** | 10.7% | 16.4% | **8.0%** |
| 200 | 6.5% | 7.8% | **4.8%** | 9.1% | 10.2% | **6.3%** |
| 500 | 6.1% | 4.6% | **2.9%** | 7.9% | 8.1% | **5.2%** |
| 2000 | 2.2% | 2.3% | **1.7%** | 3.9% | 4.7% | **4.1%** |

## H.6   Sensitivity to the guidance strength

The guidance weight $\zeta$ is a key hyperparameter in our framework, manually chosen for each task by tuning on a small validation set. This is a common practice in guided diffusion models, as the optimal weight often depends on the forward operator and prior data distribution. We provide a list of tuned $\zeta$ values in Table 4. In our experience, the model's performance is stable across wide ranges of $\zeta$. However, values that are too high can cause sampling to become unstable and diverge, while values that are too low result in weak guidance and less accurate reconstructions.

We conducted an ablation study to evaluate performance on Darcy Flow and Poisson equation forward/inverse problems across a range of $\zeta$ values, using 500 sampling steps. The results, shown in Table 10, demonstrate expected behavior.

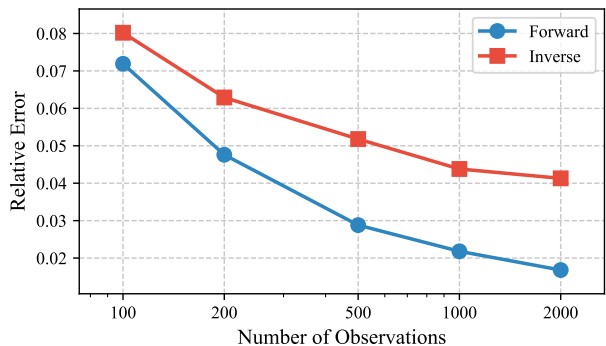

Figure 7: Comparison of the accuracy of our method with respect to the number of observations for he forward and inverse Darcy Flow problems.

Table 10: Ablation study on guidance weight $\zeta$ for Darcy Flow and Poisson problems. Errors are $L^2$ relative errors (%). NA indicates divergence due to instability.

| $\zeta$ | Darcy Forward | Darcy Inverse | Poisson Forward | Poisson Inverse |
|---|---|---|---|---|
| 1000 | 6.92% | 24.02% | 3.31% | 34.38% |
| 5000 | 2.92% | 11.01% | 2.56% | 28.05% |
| 10000 | 2.49% | 9.16% | 1.99% | 19.47% |
| 20000 | NA | 9.86% | 18.86% | 20.47% |
| 50000 | NA | 5.18% | NA | NA |

## H.7   Additional experiments on Navier-Stokes with boundary conditions

In Table 1, we present experiments for the Navier–Stokes equations with boundary conditions by using both boundary observations and $1\%$ random interior points. We also compare our methodology against a standard setup in which only $3\%$ of the data is revealed for state reconstruction. Numerical results for this comparison can be found in Table 11.

Table 11: Quantitative results of the Navier-Stokes equations with boundary conditions, where sparse observation consists of 3% of data, the same as in other PDE experiments.

| | Steps $(N)$ | Navier-Stokes with BCs | |
|---|---|---|---|
| | | Forward | Inverse |
| DiffusionPDE | 2000 | 9.78% | 4.71% |
| FunDPS (ours) | 200 | 4.62% | 3.14% |
| FunDPS (ours) | 500 | **3.48%** | **3.07%** |

## H.8   Design Choices of DiffusionPDE and FunDPS

In order to show the effectiveness of our design choice, we include an ablation study on Darcy Flow PDE by adding different components to achieve FunDPS starting from DiffusionPDE. Details are shown in Table 12, and FunDPS's superiority can be observed upon adding the components.

# I   Reproducibility of DiffusionPDE

We replicated DiffusionPDE's results using their provided code and weights. Due to reproducibility issues, we reran all experiments in communication with DiffusionPDE's authors and present the comparison in Table 13.

Table 12: Ablation study on the design choices of DiffusionPDE and FunDPS, tested on forward and inverse Darcy Flow problems. Functional diffusion noise significantly boosts capability. FunDPS achieves superior performance with fewer steps by using functional noise and neural operators.

| | Functional Noise? | Neural Operator? | Steps ($N$) | Darcy Flow | |
|---|---|---|---|---|---|
| | | | | Forward | Inverse |
| DiffusionPDE | ✗ | ✗ | 2000 | 6.07% | 14.50% |
| FunDPS w/o NO | ✓ | ✗ | 500 | 3.59% | 7.77% |
| FunDPS | ✓ | ✓ | 500 | **2.49%** | **5.18%** |

Table 13: Comparison between reported results in the DiffusionPDE paper and our reproduced results using the official code and weights.

| | Darcy Flow | | Poisson | | Helmholtz | | Navier-Stokes | | Navier-Stokes (BCs) | |
|---|---|---|---|---|---|---|---|---|---|---|
| | Forward | Inverse | Forward | Inverse | Forward | Inverse | Forward | Inverse | Forward | Inverse |
| DiffusionPDE (reported) | 2.5% | 3.2% | 4.5% | 20.0% | 8.8% | 22.6% | 6.9% | 10.4% | 3.9% | 2.7% |
| DiffusionPDE (reproduced) | 6.07% | 7.87% | 4.88% | 21.10% | 12.64% | 19.07% | 3.78% | 9.63% | 9.69% | 4.18% |

## J  Impact Statement

This research introduces FunDPS, a novel framework that significantly advances the solution of PDE-based inverse problems in function spaces. The main positive societal impact is the acceleration of scientific and engineering research by enabling more accurate and efficient modeling from sparse, noisy data in many areas.

Our released models and code are specialized for these scientific applications and are trained on simulated, non-sensitive data. Thus, they carry a low risk of societal misuse and do not produce general-purpose generative content for public consumption.

# K   Qualitative Comparison

Here we conduct qualitative comparisons between the prediction of FunDPS and DiffusionPDE on Darcy Flow (Figure 8), Poisson (Figure 9), Helmholtz (Figure 10) and Navier-Stokes (Figures 11 and 12) problems.

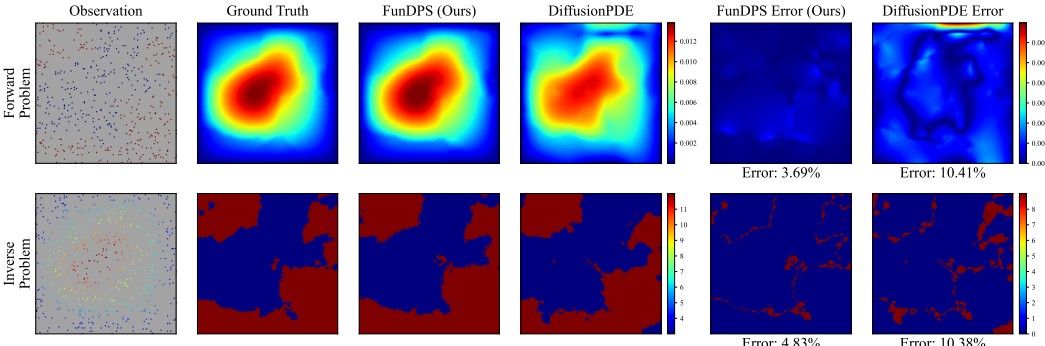

Figure 8: We compare the results of our method with the diffusion-based baseline, DiffusionPDE, on the Darcy Flow problem. The first column shows the 3% observed measurements, while the second column shows the corresponding ground truth (note that these two states are in different spaces). The middle two columns show the reconstruction results of our method and DiffusionPDE, respectively. The last two columns present the absolute error between the predictions and the ground truth. We provide relative errors for this specific test sample as well.

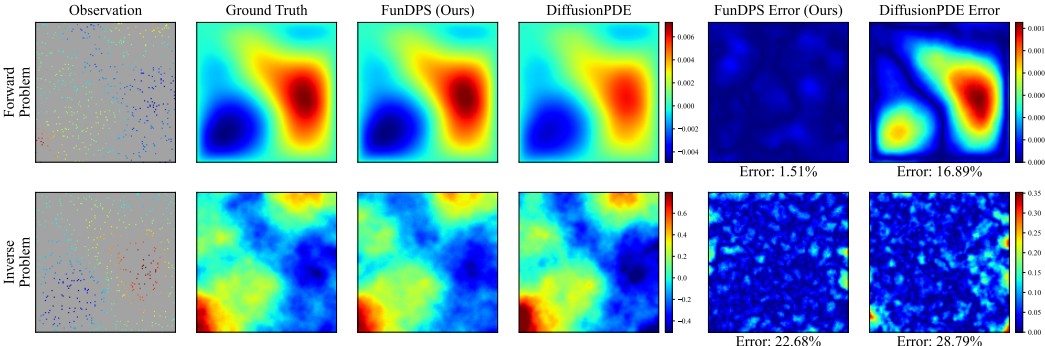

Figure 9: We compare the results of our method with the diffusion-based baseline, DiffusionPDE, on the Poisson problem. The first column shows the 3% observed measurements, while the second column shows the corresponding ground truth (note that these two states are in different spaces). The middle two columns show the reconstruction results of our method and DiffusionPDE, respectively. The last two columns present the absolute error between the predictions and the ground truth. We provide relative errors for this specific test sample as well.

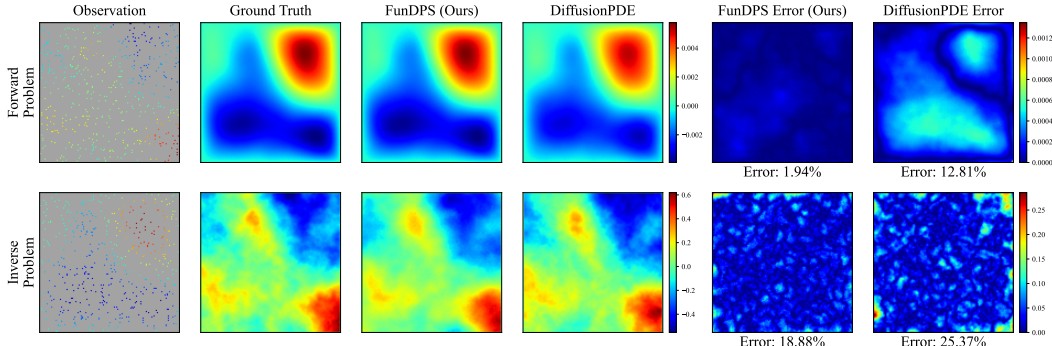

Figure 10: We compare the results of our method with the diffusion-based baseline, DiffusionPDE, on the Helmholtz problem. The first column shows the 3% observed measurements, while the second column shows the corresponding ground truth (note that these two states are in different spaces). The middle two columns show the reconstruction results of our method and DiffusionPDE, respectively. The last two columns present the absolute error between the predictions and the ground truth. We provide relative errors for this specific test sample as well.

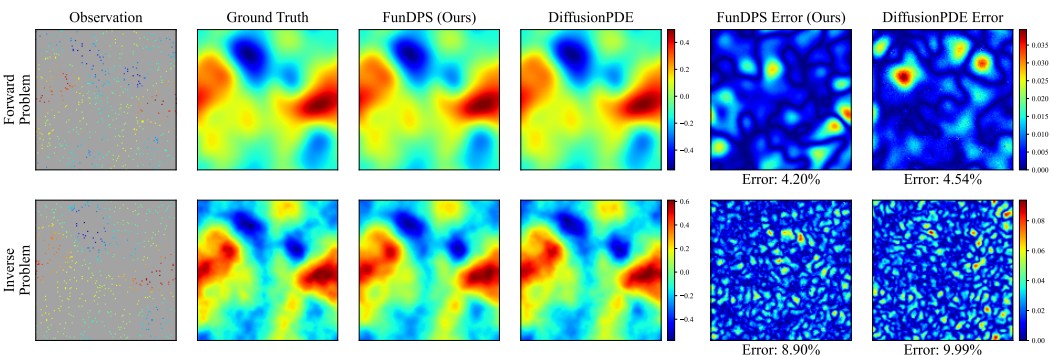

Figure 11: We compare the results of our method with the diffusion-based baseline, DiffusionPDE, on the Navier-Stokes problem. The first column shows the 3% observed measurements, while the second column shows the corresponding ground truth (note that these two states are in different spaces). The middle two columns show the reconstruction results of our method and DiffusionPDE, respectively. The last two columns present the absolute error between the predictions and the ground truth. We provide relative errors for this specific test sample as well.

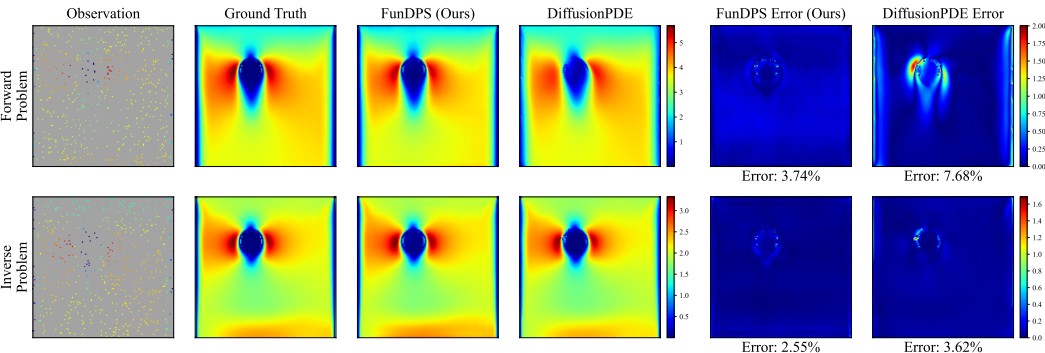

Figure 12: We compare the results of our method with the diffusion-based baseline, DiffusionPDE, on the Navier-Stokes with boundary conditions problem. The first column shows the 3% observed measurements, while the second column shows the corresponding ground truth (note that these two states are in different spaces). The middle two columns show the reconstruction results of our method and DiffusionPDE, respectively. The last two columns present the absolute error between the predictions and the ground truth. We provide relative errors for this specific test sample as well.

