# OpenReview forum: "Guided Diffusion Sampling on Function Spaces with Applications to PDEs"
_NeurIPS.cc/2025/Conference — NeurIPS 2025 poster_

### Official Review · Reviewer_dkME · 2025-06-24

**Clarity:** 2
**Significance:** 2
**Originality:** 3
**Rating:** 4
**Confidence:** 3

**Summary:**

The authors solve the problem of finding samples for a posterior function where the likelihood is given by a data fitting term and the regularization is given by a diffusion model. The setup for this problem is infinite dimensions where the parameter to be recovered belong to some functional space (rather R^n).

This types of papers are important from a theoretical point of view as they justify multiresolution work and can aid in any multiscale process.

**Questions:**

My questions are more on the practical side of the work.

Equation 13 is the approximation you use for the velocity when solving the ODE. This approximation is highly inefficient in the context of inverse problems and gradient descent. This is because if c is large (you want to actually fit the data) the ODE becomes highly stiff and you will need an implicit type method. This is why almost every PDE constrained optimization problem is solved using a Gauss-Newton type method rather than SGD. Your comparisons with other highly sub-optimal methods does not strike confidence.

**Ethical Concerns:**

["NO or VERY MINOR ethics concerns only"]

**Quality:**

3

**Strengths And Weaknesses:**

The paper may be strong theoretically although I am having a hard time following the technical content (this is more my issue that the authors). I think that researchers that are more versed in the branch of functional analysis and probability could give a more precise evaluation.
I do think that the overall content is difficult to read and work is needed to make it easier if you want a CS type publication.

I have only one question/comments (see below)

---

> ### Author Rebuttal · Authors · 2025-07-31
>
> We thank the reviewer for their time and their thoughtful, constructive feedback.
>
> > [Strengths And Weaknesses]
>
> **Answer**: We thank the reviewer for their valuable feedback and for recognizing the theoretical importance of our work. Due to the special care needed in function spaces, our formulation requires introducing notations from functional analysis.
> To enhance clarity, we will add a high-level introductory paragraph at the beginning of the Method section. Our goal is to ensure that foundational concepts are presented more simply to make the technical details accessible to a broader audience.
>
> > [Q1] Equation 13 is the approximation you use for the velocity when solving the ODE. This approximation is highly inefficient in the context of inverse problems and gradient descent. This is because if c is large (you want to actually fit the data) the ODE becomes highly stiff and you will need an implicit type method. This is why almost every PDE constrained optimization problem is solved using a Gauss-Newton type method rather than SGD. Your comparisons with other highly sub-optimal methods does not strike confidence.
>
> **Answer**: We agree that for traditional PDE-constrained optimizations, where the goal is to find a single optimal solution, the gradient descent methods are often inefficient due to stiffness. However, our approach operates in a different framework (generative posterior sampling), solving the probability flow ODE directly with an explicit and well-posed definition. Specifically, our goal is not to find a single point estimate but to draw samples from an approximate posterior distribution. This process is governed by a reverse-time ODE that integrates a powerful data-driven prior (diffusion model) with a likelihood guidance term. We would like to further clarify several points about our implementation:
> - **FunDPS is sampler agnostic**: Equation 13 defines a conditional score that provides the velocity for the reverse ODE. Since it does not prescribe any specific numerical solver nor do we use any specific solver throughout the proofs, our framework is general and can be integrated with any suitable ODE solver. The choice of sample is separate from the formulation of the guidance term (i.e. likelihood term).
> - **We use a higher-order solver, not SGD**: We do not use first-order Euler steps (analogous to not SGD). As shown in Algorithm 1, we combine the method with a second-order deterministic sampler inspired by the EDM framework [1]. This higher-order solver mitigates the instability and stiffness issues related to simpler first-order schemes.
> - **Extensive Comparisons**: During this work, we compared deep learning and GenAI baselines, including the SotA works (DiffusionPDE, FNO, PINN), and demonstrated our method's superior speed and accuracy.
>
> [1] Karras, Tero, et al. "Elucidating the design space of diffusion-based generative models." Advances in neural information processing systems 35 (2022): 26565-26577.
>
> We hope our detailed answers are helpful, and we kindly ask for a re-evaluation of our submission in light of this discussion.

---

> > ### Comment · Reviewer_dkME · 2025-08-04
> >
> > I disagree with the authors. The ODE that is solved when fitting the data well is highly stiff. I do not believe that the method has merit when data fitting is important.

---

> > > ### Author Response · Authors · 2025-08-05
> > >
> > > We appreciate your rigorous feedback. We would like to make more synchronization and clarification in response.
> > >
> > > First, we would like to clarify that our setup, which often uses a simple masking operator, is standard for many guided diffusion methods for inverse problems. If strong guidance inherently creates an intractable stiff ODE in this context, it would seem to be a general challenge for all such methods, not just our own.
> > >
> > > To that end, we would be very grateful if you could provide a reference that discusses ODE stiffness specifically in the context of guided diffusion sampling. Our understanding is that the dynamics of reverse-time SDEs/ODEs in generative sampling differ from those in classical PDE-constrained optimization, but we want to understand your perspective better.
> > >
> > > Finally, while we seek to resolve this theoretical point, we would also refer to our empirical results. Across five challenging PDE tasks, our method consistently outperforms the baselines. We conducted more comparisons and had consistent advantage with different numbers of observations (Appendix G.5 & Rebuttal to wgk8), different inverse solvers (App G.1), varying design choices (App G.7), multi-resolution training schemes (App G.3 & Rebuttal to wgk8), and additional datasets (Rebuttal to wgk8).
> > >
> > > We hope that with your guidance, we can clarify this important point about our work. Thank you again for your time.

---

> > > ### Author Response · Authors · 2025-08-08
> > >
> > > Dear Reviewer dkME,
> > >
> > > Thank you again for your time and for the insightful discussion.
> > >
> > > We are writing to respectfully follow up on our conversation regarding ODE stiffness. We are very keen to understand your perspective on this issue more deeply, as it is a crucial point for our work. To help us bridge the gap in understanding, would you be able to point us to any references that discuss ODE stiffness specifically in the context of guided diffusion sampling?
> > >
> > > Thank you for your guidance.
> > >
> > > Sincerely,
> > > Authors

---

### Official Review · Reviewer_mb5h · 2025-06-30

**Clarity:** 3
**Significance:** 2
**Originality:** 2
**Rating:** 4
**Confidence:** 3

**Summary:**

FunDPS proposes a general framework for solving PDE-based inverse problems through conditioning sampling. The core idea is to first learn the prior distribution of the underlying functions by training an unconditional diffusion model in function space. Then, conditioning on observations is achieved at inference time using a plug-and-play guidance mechanism derived from an extension of Tweedie's formula.

**Questions:**

1. How sensitive is the model's performance to the choice of $\zeta$? Please provide an ablation study showing performance across a range of $\zeta$. Moreover, is $\zeta$ manually chosen or is automatically determined under certain theoretical background?

**Ethical Concerns:**

["NO or VERY MINOR ethics concerns only"]

**Final Justification:**

All of my concerns have been addressed.
However, I will keep my score as the concerns were minor.

**Limitations:**

1. Refer to weakness.

**Quality:**

3

**Strengths And Weaknesses:**

Strengths
1. The primary contribution is the rigorous extension of diffusion models to infinite-dimensional function spaces for inverse problems.
2. The method demonstrates superior empirical performance on a variety of challenging PDE-based inverse problems, often achieving better results in fewer sampling steps than competing methods. The inclusion of discussion regarding wall-clock time addresses potential concerns about the computational overhead of calculating the guidance term's gradient.
3. The proposed multi-resolution training and inference strategy is a key practical advantage, enabling efficient handling of functions at different levels of detail and likely contributing to the model's overall performance and speed.

Weakness
1. While the paper's contributions are significant, the core components are not entirely novel in isolation. The concept of function-space diffusion models builds upon prior work (e.g., Score-based Diffusion Models in Function Space), and plug-and-play guidance is a well-established technique in finite-dimensional diffusion models.

---

> ### Author Rebuttal · Authors · 2025-07-31
>
> We thank the reviewer for their time and their thoughtful, constructive feedback.
>
> > [W1] While the paper's contributions are significant, the core components are not entirely novel in isolation. The concept of function-space diffusion models builds upon prior work (e.g., Score-based Diffusion Models in Function Space), and plug-and-play guidance is a well-established technique in finite-dimensional diffusion models.
>
> **Answer**: Thank you for your comment. We agree that our work builds upon important prior research. However, our primary contribution is based on a novel integration of these ideas. Most importantly, FunDPS provides essential theoretical and practical contributions to make this combination work and be effective for solving ill-posed inverse problems arising in PDEs.
> While plug-and-play guidance was established in finite dimensions, a rigorous theoretical framework in the infinite-dimensional setting of function spaces was absent. Our work introduces the first generalization of Tweedie’s formula to Banach spaces. This is a key novelty that enables us to propose principled gradient-based guidance for diffusion models operating directly on a function. It goes beyond a simple combination of existing works, enabling design choices that are crucial for superior accuracy, speed, and multi-resolution strategies.
>
>
> > [Q1] How sensitive is the model's performance to the choice of \zeta? Please provide an ablation study showing performance across a range of \zeta. Moreover, is \zeta manually chosen or is automatically determined under certain theoretical background?
>
> **Answer**: This is an excellent question regarding the key hyperparameter in our framework. Currently, the guidance weight $\zeta$ is manually chosen for each task by tuning on a small validation set. This is a common practice of guided diffusion models, as optimal weight often depends on the forward operator and prior data distribution. We provide a list of tuned $\zeta$ values used for all five PDE tasks in Appendix F, Table 4. In our experience, the model’s performance is stable in wide ranges. However, values that are too high can cause the sampling to become unstable and diverge, while the values that are too low result in weak guidance and less accurate reconstructions.
>
> Note that after several trials, we follow ideas from the DPS paper, and we use adaptive guidance weighting based on noise level as highlighted in Appendix F. We agree that developing a method to automatically determine $\zeta$ is an important direction for future research, which could help overcome inherent challenges in this field.
>
> Finally, we followed the suggestion and conducted an ablation study to show the performance of Darcy and Poisson equation forward/inverse problems across a range of $\zeta$ values. We can observe that FunDPS has expected behaviour across different ranges.
> |zeta|Darcy Forw.|Darcy Inv.|Poisson Forw.|Poisson Inv.|
> |-|-|-|-|-|
> |1000|6.92|24.02|3.31|34.38|
> |5000|2.92|11.01|2.56|28.05|
> |10000|**2.49**|9.16|**1.99**|**19.47**|
> |20000|NA|9.86|18.86|20.47|
> |50000|NA|**5.18**|NA|NA|
>
> We hope our detailed answers are helpful, and we kindly ask for a re-evaluation of our submission in light of this discussion.

---

> > ### Comment · Reviewer_mb5h · 2025-08-04
> >
> > 1. Thanks authors for the response. My concerns have been addressed.
> >
> > 2. thanks authors for the additional experiments and corresponding analysis. Please include this in the final version if accepted.

---

> > > ### Author Response · Authors · 2025-08-05
> > >
> > > Thank you for your positive feedback on our submission and rebuttal. We are glad the additional experiments were helpful, and we will include them and the related discussion in the final version. We appreciate your time and constructive advice :)

---

### Official Review · Reviewer_uwzC · 2025-07-03

**Clarity:** 3
**Significance:** 2
**Originality:** 3
**Rating:** 4
**Confidence:** 4

**Summary:**

The papers propose a general framework for conditional sampling in PDE-based inverse problems, to recover the whole solutions from extremely sparse or noisy measurements, by a function-space diffusion model and plug-and-play guidance for conditioning.

**Questions:**

Please see the weaknesses for specific questions.

**Ethical Concerns:**

["NO or VERY MINOR ethics concerns only"]

**Final Justification:**

The author provides extensive rebuttal to reply my questions, which are helpful to address most concerns. See comments for more details. Considering this, I am pleased to raise my score.

**Limitations:**

Yes, the author discusses the limitations of the proposed method in the conclusion.

**Quality:**

3

**Strengths And Weaknesses:**

Strengths:

+ This paper is well written and easy to follow.

+ The investigated problem is well-motivated to design the diffusion model in the function space for solving PDE problems.

+ The proposed method looks reasonable to achieve the goal, with promising results achieved compared with previous baseline such as DiffusionPDE.


Weaknesses:

- The key contribution lies in the directly extending DiffusionPDE to the function space through function representation. The contribution on the multi-resolution training and inference for acceleration, and conditional sampling algorithm for solving PDE problems may be limited, since they are mostly integrated from previous works such as DiffusionPDE.

- The results with different inverse solvers in Table 5 are interesting. The results show that DAPS and DDNM is far worse than the DPS, which may diverge from the results in previous works. It would be helpful to elaborate what could be the underlying reason, like due to the function space formulation or PDE task?

- A key baseline compared in the paper is the DiffusionPDE, with the main difference in finite v.s. infinite representation space for training diffusion models, with claimed key contribution in extended Tweedie’s Formula. But another possible simpler approach is to represent measurement in the function space and inject to each diffusion sampling step to avoid estimating the Tweedie’s Formula. This was introduced in the diffusion inverse solvers before the introduction of DPS. Would this be possible to solve the target problem?

- The paper claims the sampling from posterior distribution with DPS guidance, while DPS is shown not to provably sample from posterior, such as in [1]. Can the author elaborate more on how to guarantee the posterior sampling as claimed?

[1] Cardoso, G., Janati El Idrissi, Y., Le Corff, S., & Moulines, É. (2024). Monte Carlo guided Denoising Diffusion Models for Bayesian Linear Inverse Problems. In Proceedings of ICLR 2024

---

> ### Author Rebuttal · Authors · 2025-07-31
>
> We thank the reviewer for their time and their thoughtful, constructive feedback.
>
> > [W1] The key contribution lies in the directly extending DiffusionPDE to the function space through function representation. The contribution on the multi-resolution training and inference for acceleration, and conditional sampling algorithm for solving PDE problems may be limited, since they are mostly integrated from previous works such as DiffusionPDE.
>
> **Answer**: Thank you for this critical question. We would like to clarify the distinction and novelty of our work compared to DiffusionPDE. While we use their datasets for a fair and direct comparison, our methodology is fundamentally different. Shortly speaking, the only design choice we borrowed from them is the joint-embedding mechanism. In other aspects, FunDPS has important contributions as follows:
> - **A New Discretization-Agnostic Framework**: Our core contribution is a framework that is inherently discretization-agnostic. FunDPS achieves this by using a U-shaped Neural Operator (U-NO) and Gaussian Random Field (GRF) noise model, which operates on continuous function representations. This differs sharply from DiffusionPDE, which is based on convolutional networks and is fundamentally tied to a fixed grid resolution (Please check Table 2 in the response to reviewer wgk8). This is not only a change in representation, but a critical shift in the underlying model that leads to inherent resolution-invariance, which was the key limitation of prior works. **Our ablation study in Table 9 further confirms the benefits of function-space approach** with significant performance boost.
> - **Novel Theoretical Guarantees in Function Spaces**: To provide principled guided sampling on the infinite-dimensional settings, we propose and prove the generalization of Tweedie’s formula to Banach spaces. This theoretical result is a critical contribution that provides a mathematical relation between the score function and the denoising operator in function spaces, which is crucial for many plug-and-play guidance mechanisms, including ours.
> - **Multi-resolution Capabilities and State-of-the-Art Performance**: Our function space formulation inherently leads to powerful multi-resolution training and inference strategies that were impossible with prior works as they are tied to fixed resolutions. For instance, **our ReNoise inference pipeline achieves 25x wall-clock speedup** over DiffusionPDE. Furthermore, our resulting framework obtains 32% average accuracy improvement with 4x fewer sampling steps.
>
> We will revise the introduction to more explicitly highlight these fundamental distinctions.
>
>
> > [W2] The results with different inverse solvers in Table 5 are interesting. The results show that DAPS and DDNM is far worse than the DPS, which may diverge from the results in previous works. It would be helpful to elaborate what could be the underlying reason, like due to the function space formulation or PDE task?
>
> **Answer**: We thank the reviewer for highlighting this interesting result. The significant underperformance of DDNM and DAPS in our setting is primarily due to the extreme sparsity of observation data, not the function space formulation, which we found after extensive investigation.
> - DDNM relies on projecting the sample onto measurement subspace at each step. With only 3% of observation points, the measurement subspace is extremely low-dimensional compared to the overall function space. As a result, the projection provides very weak guidance and leads to poor reconstruction.
> - DAPS uses a decoupled noise annealing schedule. The core issue comes from the localness of guided updates during Langevin dynamics. Only the points with observation are updated and others are just added with noise. This makes the intermediate state after Langevin dynamics discontinuous and out-of-distribution, which hinders performance. If we increase the # of sampling steps to 20,000, the accuracy can match DPS, but with 100x more time. The same trend is also observed in image-domain frameworks as we provide the quantitative results.
>
> We have figures available to back our findings. We will include them in the camera-ready version and hope that it can provide insights into the advantages and disadvantages of different inverse solvers.
>
>
> > [W3] A key baseline compared in the paper is the DiffusionPDE, with the main difference in finite v.s. infinite representation space for training diffusion models, with claimed key contribution in extended Tweedie’s Formula. But another possible simpler approach is to represent measurement in the function space and inject to each diffusion sampling step to avoid estimating the Tweedie’s Formula. This was introduced in the diffusion inverse solvers before the introduction of DPS. Would this be possible to solve the target problem?
>
> **Answer**: Thank you for this insightful question about alternative approaches. To ensure we understand accurately, we would be grateful if the reviewer could clarify which specific prior works they have in mind. This may refer to resampling-based or projection-based methods. Proceeding with that assumption, we can explain why such an approach, while powerful in some domains, is ill-suited for the PDE inverse problems here.
> - **Instability of forward operators on noisy states**: Representing measurement would often require applying the forward operator $A$ (i.e. masking or PDE solver) to a noisy intermediate state $a_t$ to enforce consistency. PDE solvers are often ill-posed or numerically unstable when their input is a noisy, non-physical function, which is the case in huge noise spaces. FunDPS prevents this by first using denoiser to predict a clean state $\hat{a_0} = D_\theta(a_i, t_i)$, and then applying the forward operator to this physically plausible estimate $A(\hat{a_0})$. This leads to more stable gradient guidance.
> - From a practical standpoint, **our generalized Tweedie's formula is computationally inexpensive**. Plus, both methods involve approximations, so there is no clear advantage of one over another.
>
> With that said, our generalization of Tweedie’s formula bridges the denoising target and the score function required for guidance. We believe this contribution is not just crucial for our method but can also empower a wide range of future guided diffusion solvers that operate in function spaces.
>
>
> > [W4] The paper claims the sampling from posterior distribution with DPS guidance, while DPS is shown not to provably sample from posterior, such as in [1]. Can the author elaborate more on how to guarantee the posterior sampling as claimed?
>
> **Answer**: Thank you for this important point and for providing the reference. The reviewer is correct that our method, like other guided diffusion techniques, provides an approximation of true posterior distribution. The approximation arises because the time-dependent log-likelihood term in the conditional score (Equation 6) is intractable, and we approximate using the conditional expectation of the clean sample (Equation 9 & 10).
> We want to highlight that **even in standard finite dimensions, there is no inference-time inverse solver that has guarantees for true posterior (w/o training)**. Methods, such as Sequential Monte Carlo (SMC, TDS, etc.), have stronger *asymptotic* guarantees, but they are computationally more intense. It would be interesting to extend such methods to function spaces in future work.
> We will revise our manuscript to claim “approximate posterior sampling”. We will further add a sentence in Section 3.3 to explicitly state the source of this approximation by referring to the suggested paper [1].
>
> [1] Cardoso, G., Janati El Idrissi, Y., Le Corff, S., & Moulines, É. (2024). Monte Carlo guided Denoising Diffusion Models for Bayesian Linear Inverse Problems. In Proceedings of ICLR 2024
>
> We hope our detailed answers are helpful, and we kindly ask for a re-evaluation of our submission in light of this discussion.

---

> > ### Comment · Reviewer_uwzC · 2025-08-05
> >
> > Thank the author for the thorough response to answering my questions. Most of my concerns have been well addressed. For W3, the author is correct what I was thinking about is the projection-based approach which requires applying the forward operator. Thank the author for pointing out the potential issue of this solution due to the limitation of PDE solvers, which sounds reasonable. For W4, I agree there is a trade-off between computational efficiency and the guarantee of the posterior sampling. I just would like to use [1] to show that for DPS based solution, it is well recognized that the samples will not be guaranteed from posterior since [1] directly show the results comparison in synthetic toy examples (the author does not need to bother to cite this work or not beyond the discussion). But it may be important to clarify the claim in the paper since the proposed method would also have the same limitation due to fundamental algorithm design, so the resulted solution may fall off the true posterior distribution? Considering the extensive rebuttal provided by the author, I am pleased to raise my score.

---

> > > ### Author Response · Authors · 2025-08-05
> > >
> > > Thank you for your detailed follow-up and for raising your score. We appreciate your time and are pleased that we were able to address most of your concerns.
> > >
> > > Regarding the guarantee of posterior sampling, it is an important point! We believe this is a beneficial discussion for readers to have a clear perception of our method and guided diffusion approaches in general. We will revise the final version of our paper to bring up this limitation and properly frame our contribution in this context.
> > >
> > > Thank you again for your constructive review.

---

### Official Review · Reviewer_8pVy · 2025-07-03

**Clarity:** 3
**Significance:** 3
**Originality:** 3
**Rating:** 5
**Confidence:** 3

**Summary:**

The paper focusses on solving PDE-based inverse problems formulated in infinite dimensions using conditional diffusion processes. They formulate an infinite-dimensional conditional score via a density with respect to a Gaussian measure. They show that this can be decomposed into a likelihood term and prior or unconditional score. The likelihood is approximated in closed form by approximating the conditional reverse diffusion process with its conditional expected value. This expected value is proved to satisfy an infinite-dimensional Tweedie’s formula. Using the resulting closed form approximation of the guidance alongside a trained (unconditioned) score operator allows one to obtain samples from the posterior. The method is compared experimentally to other methods on forward and inverse problems for different PDE tasks.

**Questions:**

For Tweedie’s formula, is there anything that can be said about the assumption that the score of $\nu$ is Fréchet differentiable? When will this hold?

**Ethical Concerns:**

["NO or VERY MINOR ethics concerns only"]

**Final Justification:**

I think the paper has strong theoretical results (formulating the conditional diffusion problem to function spaces, and an infinite-dimensional Tweedie's formula) coupled with good experimental results.

**Limitations:**

Yes.

**Paper Formatting Concerns:**

None.

**Quality:**

4

**Strengths And Weaknesses:**

Strengths:

1. In general, I find the submission to be well-written.

2. The method is compared experimentally to well-known PDE solvers across five different PDE experiments, which are the same as those considered in DiffusionPDE. The method performs well on these baselines, outperforming the other methods whilst using a quarter of the time discretisation steps used in DiffusionPDE.

3. The submission proves an infinite-dimensional Tweedie’s formula. Tweedie’s formula is an important result used in finite-dimensional diffusion models. I find its extension to infinite-dimensions to be a valuable result.

4. The paper also details a new training technique for learning the prior distribution at a lower cost using multi-resolution training. This is coupled with what they term multi-resolution inference also accelerating the inference time per step. I think this is a valuable contribution which can also be used in the infinite-dimensional unconditional problem.


Weaknesses:


1. In order to obtain a closed form expression for the likelihood, the conditional measure of the reverse diffusion process is approximated with its expectation. Can you comment on the error induced by this approximation?

2. The paper introduces the conditional score of $\nu^u_t$ as the object of interest, however it’s not clear to me exactly how it is used in the given setup for sampling. I think it would be beneficial to make this explicit, for example, by giving the time-reversal of (4) in terms of the conditional score.

3. The submission introduces a lot of different notations for various measures and conditional distributions. I think this is necessary, however I would find a paragraph on the notation useful to refer back to, containing the main measures e.g. $\mu, \eta, \gamma, \nu$.


4. The following weaknesses are minor and do not affect my score.

    - In line 239 “firt” should be “first”.

    - In Equation (13) should the scalar be $\frac{c^2}{2}$ or $\frac{c}{2}$? Equation (11) uses $C_{\eta}^{-1/2}$, which according to line 277 should be given by $c^{1/2}I.$


In summary, I find the paper gives an important and novel contribution in extending diffusion models for Bayesian inverse problems into function space. Experimentally, the method is shown to work well in comparison to existing methods and I recommend accepting this paper. However, I think there are some presentational issues which if fixed would improve the paper. Most importantly, making the infinite dimensional sampling scheme clearer.

---

> ### Author Rebuttal · Authors · 2025-07-31
>
> We thank the reviewer for their time and their thoughtful, constructive feedback.
>
> ## W1
> **Problem**: In order to obtain a closed form expression for the likelihood, the conditional measure of the reverse diffusion process is approximated with its expectation. Can you comment on the error induced by this approximation?
>
> **Answer**: This is an important question and we thank the reviewer for pointing it out. In the Euclidean setting, the error of this kind of approximation is described precisely by the Jensen gap of the conditional density, i.e., $E_x [p(y|x)] - p(y|E[x])$. Depending on the form of the noise, an upper bound on this gap can be given [1].  We believe a similar result will hold in our setting, where the notion of a density is now replaced by an appropriate notion of a Radon-Nikodym derivative. However, further research is needed for a formal proof, and we plan to spend more effort in this direction.
>
> [1] Chung, Hyungjin, et al. "Diffusion posterior sampling for general noisy inverse problems." arXiv preprint arXiv:2209.14687 (2022).
>
> ## W2
> **Problem**: The paper introduces the conditional score of \nu_t^u as the object of interest, however it’s not clear to me exactly how it is used in the given setup for sampling. I think it would be beneficial to make this explicit, for example, by giving the time-reversal of (4) in terms of the conditional score.
>
> **Answer**: The conditional score of $\nu_t^u$ can be informally thought of as $\nabla_{a} p_t(a_t \mid u)$, i.e., the usual notion of a score function conditioned on an additional variable $u$. One of the main contributions of our work is an approximation of this conditional score in function spaces using an unconditional score and an additional guidance term (Eqn 13). To sample using this score, one starts by initializing $a_T$ as a Gaussian random field, and iteratively moving in the direction of this conditional score (Eqn 14 and L3-11 in Alg 1).
>
> We agree with you that this could be explained more precisely in our submission. Your suggestion of writing the time reversal of (4) in terms of this conditional score will make this more clear to readers, and we will update our paper to include this.
>
>
> ## W3
> **Problem**: The submission introduces a lot of different notations for various measures and conditional distributions. I think this is necessary, however I would find a paragraph on the notation useful to refer back to, containing the main measures e.g. \mu, \eta, \gamma, \nu.
>
> **Answer**: We thank the reviewer for their constructive feedback. In the final manuscript, we will include a more comprehensive notation paragraph to enhance the paper’s readability.
>
>
> ## W4
> **Problem**:
> - In line 239 “firt” should be “first”.
> - In Equation (13) should the scalar be c^2/2 or c/2? Equation (11) uses C_\eta^{-1/2}, which according to line 277 should be given by c^{1/2}I.
>
> **Answer**: We thank the reviewer for reading the paper thoroughly and finding these typos. Indeed, both points are valid and fixed in the paper. For the second point, we will fix Equation (13) to be $\frac{c}{2}$. As we note in lines 286-288, the constant term is $\frac{c}{2}$ (not $\frac{c^2}{2}$) and absorbed into the final guidance weight hyperparameter $\zeta$. This hyperparameter has been tuned for each specific problem based on a small validation set (Appendix F).
>
>
> ## W*
> **Problem**: In summary, I find the paper gives an important and novel contribution in extending diffusion models for Bayesian inverse problems into function space. Experimentally, the method is shown to work well in comparison to existing methods and I recommend accepting this paper. However, I think there are some presentational issues which if fixed would improve the paper. Most importantly, making the infinite dimensional sampling scheme clearer.
>
> **Answer**: Thank you very much for the positive feedback and constructive suggestions. We will revise the final manuscript based on all valuable reviews we get to make the main idea clearer. The infinite-dimensional sampling scheme is one of our core contributions, and we will enhance the content by rethinking definitions, better organization, and additional discussions.
>
> ---
>
> ## Q1
> **Problem**:  For Tweedie’s formula, is there anything that can be said about the assumption that the score of \nu is Fréchet differentiable? When will this hold?
>
> **Answer**: This is a good question that touches on a subtle theoretical point. The answer depends on the regularity of the data distribution $\mu$.
> The Fréchet differentiability of the score of $\nu$ is fundamentally tied to the smoothness of the Radon-Nikodym derivative $\frac{d\nu}{d\gamma}$, where $\gamma$ is the base Gaussian measure.
>
> For a Gaussian measure $\gamma$, the natural space of directions for differentiation is its Cameron-Martin space. Since our perturbed measure $\nu$ is equivalent to $\gamma$ (i.e., they are mutually absolutely continuous, per Prop B.2), $\nu$ inherits much of the structure of $\gamma$. Therefore, assuming the score of $\nu$ is Fréchet differentiable with respect to perturbations from the Cameron-Martin space is a reasonable starting point, provided the data distribution $\mu$ is sufficiently regular. Without further assumptions on $\mu$, however, a more general statement is difficult. For a complete technical discussion, we refer to [2, Chapter 5].
>
> One special case, though, where it is possible to give a definitive answer is when the data distribution $\mu$ is itself Gaussian. In this scenario, the perturbed measure $\nu$, being the convolution of two Gaussians, is also a Gaussian measure, and thus its Frechet differentiability is fully characterized by its Cameron-Martin space. See [3, Example 1] for a full proof.
>
> [2] Bogachev, Vladimir Igorevich. Gaussian measures. No. 62. American Mathematical Soc., 1998.
>
> [3] Lim, Jae Hyun, et al. "Score-based diffusion models in function space." arXiv preprint arXiv:2302.07400 (2023).

---

> > ### Comment · Reviewer_8pVy · 2025-08-04
> >
> > Thank you for your detailed answers. After reading the reviews and rebuttals, I think this is a nice paper with strong theoretical and experimental contributions and continue to recommend its acceptance.

---

> > > ### Author Response · Authors · 2025-08-05
> > >
> > > Thank you for your thoughtful review and positive feedback. We are very grateful for your support and are delighted that you found our contributions to be strong. We sincerely appreciate your time.

---

### Official Review · Reviewer_wgk8 · 2025-07-04

**Clarity:** 3
**Significance:** 2
**Originality:** 3
**Rating:** 3
**Confidence:** 3

**Summary:**

The paper introduces FunDPS, a novel framework for solving inverse problems involving partial differential equations (PDEs) using diffusion models in function spaces. It combines a discretization-agnostic denoising diffusion model with a plug-and-play guidance mechanism that uses sparse observations for conditional sampling. FunDPS achieves state-of-the-art accuracy and speed, demonstrating significant improvements over prior methods in recovering solutions from as little as 3% of observed data.

**Questions:**

(1) What are the main intuition of generalizing the Tweedie's formula to the function space?
(2) When you compare your method with DiffusionPDE, are you using the same neural network backbone architecture?
(3) What are the challenges to generalize this algorithm to more practical diffusion usecase like image generation?
(4) Do we need to apply any assumption (theoretical or practical) to this inverse problem formulation if we want to extend Tweedie's formulation to infinite-dimensional Banach spaces?

**Ethical Concerns:**

["NO or VERY MINOR ethics concerns only"]

**Final Justification:**

I hold my opinion of borderline reject as I was not convinced of the necessity of going beyond euclidean space (also skeptical of the euclidean space experiment results). But I am open to any final paper decision.

**Quality:**

3

**Strengths And Weaknesses:**

Strengths: The paper adopts a function space formulation of diffusion and can achieve faster and better sampling from experiment. The paper has a clear explanation of the theoretical innovations and is well written. The paper proposes a novel discretization-agnostic framework which can generalize across different resolution

Weakness:
(1) My main concern is incremental innovation compared to DiffusionPDE, although theoretically fancy, Alorithm 1 has exactly the same procedure as DiffusionPDE, so could you point out what is the main difference with diffusionPDE from the implementation perspective? also why the PDE loss is not included in the loss term? Is it a negative contribution to both DiffusionPDE and FunDPS? If it is, it will be more fair to compare both method without this term together.
(2) It will be better to highlight more the intuition and motivation to introducing the functional space especially for this PDE problem, the current version reads more like motivation for diffusion inverse model + functional space diffusion..

(3) Since the backbone is relatively small scale and influence of hyperparameters will be more obvious, it will be good to add more ablations on the influence of hyperparameters.

(4) Also will be more fair if you can perform careful hyperparameter on the baseline method for different problems as well, especially the error you report here as baseline is much higher than the original numbers reported in the DiffusionPDE paper.

(5) It will be good to include more diffusion-based PDE solver besides DiffusionPDE, also the model 54M is relatively small, not sure if it can generalize well on more general or ood test datasets.

(6) Current experiment applies 3\% observation and will be interesting to see how the method along with diffusionPDE can handle fewer observation and what are the lower limit of observation of diffusion-based approach.

(7) I did not fully understand the multi-resolution training, but want to know will this training method influence the sampling steps, final accuracy, and does using coarser grid break the 3\% observation assumption? If this multi-resolution training will influence final accuracy, are you applying it to the diffusion-based baselines as well?

(8) As in DiffusionPDE paper which compares DiffusionPDE with Diffusion with CFG, does it make sense to compare FunDPS with the functional-space diffusion with CFG as well?

---

> ### Author Rebuttal · Authors · 2025-07-31
>
> We thank the reviewer for their time and their thoughtful, constructive feedback.
>
> > [W1] Concern about incremental innovation compared to DiffPDE. Plus, is PDE loss a negative contribution to DiffPDE and FunDPS?
>
> A: Thanks for the question. Inside the sampling loop, our work offers three key improvements over DiffPDE:
> - **Architecture**: Our core contrib. is a discretization-agnostic generation framework,  by using U-NO and GRF noise model operating on continuous representations. However, DiffPDE, which is tied to a fixed grid resolution, cannot generalize to different resolutions well. **This generalization is important because in scientific domains, data are often collected in different resolutions.** Our ablation study (Tab 9) further confirms the benefits of function-space approach.
> - **Theory**: We introduce and prove the first generalization of Tweedie’s formula to Banach spaces, a critical theoretical guarantee for guided sampling in function spaces, to provide principled guided sampling when using GRF noise.
> - **Multi-resolution Capabilities and SotA Performance**: These contributions unlock new multi-resolution strategies, achieving a 25x inference speedup and a 32% accuracy improvement.
>
> As for the second question, PDE loss **is** included. It is expressed in a uniform way with the obs. loss. Contrary to the concern, both our work (App F) and DiffPDE (Sec 4.7) demonstrate that PDE loss improves accuracy.
>
> > [W2] Highlight the intuition and motivation to introducing the functional space, especially for this PDE problem.
>
> A: Thank you for suggestion. Our main intuition is that physical systems described by PDEs are inherently continuous, and numerical solutions are discretizations of these dynamics. One major challenge in scientific computing is that a model trained on one specific grid often fails to generalize to another. Our function-space formulation addresses this by learning the resolution-agnostic mapping between the parameter and solution functions. This approach is more natural and well-suited as the real-world data inherently comes at varying resolutions. It is also becoming standard in AI4Sci community. Furthermore, this unlocks significant efficiency gains from our multi-resolution training and inference strategies.
>
> > [W3] It will be good to add more ablations on the influence of hyperparameters.
>
> A: We agree that selecting hyperparameters is important. In our submission, we analyze various factors: # of observations (App G.5); different inverse solvers (App G.1); different training schemes (App G.3); ratio of low resolution sampling steps (App G.4); and varying design choices (App G.7). Notably, our model has a wide stable range for guidance weights, shown in the rebuttal to mb5h.
>
> > [W4] It will be more fair to perform careful hyperparameter on the baseline, especially the error you report here is much higher than the original numbers.
>
> A: Thanks for the important question. To ensure a fair comparison, we used the official implementation and weights from the DiffPDE authors. We were in direct communication with them, and they confirmed that the results are the same as they could reproduce. In fact, for Helmholtz and NS1 cases, the reproduced baseline is even stronger than the original. We're confident this provides the most rigorous foundation for comparison.
>
> > [W5] It will be good to include more diffusion-based PDE solver. Also the model 54M is relatively small, not sure if it can generalize well on more general datasets.
>
> A: Thank you for your suggestion. We have included an exp. in App G.1 that evaluates three plug-and-play inverse solvers.
> Regarding model scale, we chose a 54M parameter model to make a fair comparison with DiffPDE. In ablation, the largest model has 180M parameters, comparable to the typical diffusion model size of 300M for ImageNet64.
>
> To further confirm generalizability, we provide additional exp. on real-world geophysical Darcy dataset, where we outperform all tuned baselines. The dataset is generated following [1].
> ||Forw.|Inv.|
> |-|-|-|
> |FunDPS|2.6|2.6|
> |DiffPDE-2000|3.0|3.9|
> |FNO|5.85|7.29|
> |U-NO|5.53|6.92|
> |U-FNO|3.12|3.80|
>
> Finally, we agree that scaling to larger models and more complex datasets is an exciting future direction.
>
> [1] Wen, Gege, et al. "U-FNO—An enhanced Fourier neural operator-based deep-learning model for multiphase flow."
>
> > [W6] How the method along with DiffPDE can handle fewer observation and what are the lower limit?
>
> A: Good question! We have investigated this in App G.5. We further conducted exp. with DiffPDE, shown below. Our method consistently outperforms across varying # of observations. As for the lower limit, it really depends on the use case; it maintains under 10% error with as little as 0.5% obs.
> |||Forw.|||Inv.|||
> |-|-|-|-|-|-|-|-|
> |# of Obs|DiffPDE-2000|DiffPDE-500|FunDPS|DiffPDE-2000|DiffPDE-500|FunDPS|
> |100|7.2|13.8|7.2|10.7|16.4|8.0|
> |200|6.5|7.8|4.8|9.1|10.2|6.3|
> |500|6.1|4.6|2.9|7.9|8.1|5.2|
> |2000|2.2|2.3|1.7|3.9|4.7|4.1|
>
> > [W7] Will multi-resolution training influence sampling steps and accuracy? Does using coarser grid break the 3% observation assumption? Are you applying it to the diffusion-based baselines as well?
>
> A: i) The number of inference steps and guidance weight are not affected by the strategy because our model can generalize.
>
> ii) The observation is properly handled. 3% observation means 125 and 500 observed pts for 64 and 128 resolution.
>
> iii) The multi-resolution inference cannot be applied to UNet as it cannot process different resolutions without training. The training strategy can be applied to DiffPDE. As shown below, FunDPS still outperforms baseline by a large margin with any training strategy. Notably, the UNet trained on 64x64 performs very badly on 128x128.
> |||Darcy Forw.|||
> |-|-|-|-|-|
> |Training|Inference|DiffPDE-2000|DiffPDE-500|FunDPS|
> |64|64|6.38|7.96|3.03|
> |64|128|33.72|35.57|3.64|
> |128|128|6.07|4.60|2.74|
> |Mixed|128|3.83|4.53|2.49|
>
> > [W8] Compare FunDPS with the functional-space diffusion with CFG?
>
> A: Thanks for this insightful comment. However, we believe such comparison is out of scope for the current work because
> - Adapting CFG to function space is a non-trivial task and could be a new paper in itself. Our focus is on a principled posterior sampling framework with a plug-and-play solver.
> - We compare primarily against DiffPDE, as it was already shown to outperform CFG models in its original paper. This provides a more informative evaluation against a stronger baseline.
>
> > [Q1] The main intuition of generalizing the Tweedie's formula to the function space?
>
> A: At a high level, Tweedie’s formula states that if $Y = X + Z$ is a noisy observation of a random variable $X$ corrupted by Gaussian noise $Z \sim N(0, C)$, then the best guess (i.e. cond. expectation) for the value of $X$ given a noisy obs $Y=y$ is given by $E[X|Y=y]=y+C\nabla_y\log p(y)$. In other words, we take the observation $y$ and correct it by adding the vector $\nabla_y\log p(y)$, moving $y$ in the direction which most quickly increases its likelihood. The additional $C$ term accounts for correlations in the Gaussian noise.
>
> When generalizing this to function spaces, the intuition remains the same, with the key difference being $X,Y$ are now random functions, and $Z$ corresponds to a Gaussian random field. This means that the gradient must be replaced with Frechet derivative, which is no longer a vector but rather a function itself. Moreover, $C$ is no longer a covariance matrix, but rather a covariance operator which acts on functions. The shape of this corrective function is determined by the functional derivative of the log-probability density, smoothed by the covariance operator of the noise process.
>
> However, a key theoretical difference in function spaces is that probability densities only exist under certain strict conditions. To ensure a well-defined density exists in function space, the Cameron-Martin theorem (B.1) gives us the precise conditions, effectively requiring our data to live in a specific subspace (the Cameron-Martin space) where the signal and noise measures “agree” with each other.
>
> We will update Sec 3.2 for the sake of more clarity.
>
> > [Q2] Compared with DiffPDE, are you using the same neural network arch?
>
> A: No, FunDPS uses a U-shaped Neural Operator, which is designed to be resolution-agnostic. This is the one aspect of our contribution.
>
> > [Q3] What are the challenges to generalize this algorithm to more practical diffusion usecase like image generation?
>
> A: Thanks for great question. PDE problems and image generation are important usecases. Generalizing function-space methods to image domain is an active research area. A key challenge is that NOs can oversmooth the sharp features in natural images. In principle, our framework could be adapted, but it is beyond the scope of this paper.
>
> > [Q4] Do we need to apply any assumption to this inverse problem formulation if we want to extend Tweedie's formulation to infinite-dimensional Banach spaces?
>
> A: Yes, the core assumption is that the data is supported on the Cameron-Martin space of the noise process. We refer to our Thm B.6.
> This assumption is the infinite-dimensional equivalent of ensuring a well-defined probability density exists, which is a prerequisite for defining the score function at the heart of Tweedie's formula. As we noted, it is a standard requirement in function-space diffusion literature.
> While verifying this assumption theoretically can be difficult, a common workaround is to pre-smooth the data, for instance by applying the operator $C^{1/2}$ (the square root of the noise covariance). This effectively projects the data onto the Cameron-Martin space, meeting the assumption. For more details on this technique, we point to [1, Sec 4.2].
>
> [1] Lim, Jae Hyun, et al. "Score-based diffusion models in function space." (2023)
>
> We hope our detailed answers are helpful, and we kindly ask for a re-evaluation after this discussion.

---

> > ### Author Response · Authors · 2025-08-06
> >
> > We hope that our response has adequately addressed the reviewer's concerns. Before the end of discussion period, we are kindly asking the reviewer whether their concerns have been resolved or if there are additional ways in which we can improve the paper. Thank you.

---

> > ### Comment · Reviewer_wgk8 · 2025-08-08
> >
> > I thank the authors for their detailed response, which clarifies most of my concerns. I hope the authors can incorporate some of the feedback in the revised version.

---

> > > ### Author Response · Authors · 2025-08-08
> > >
> > > Dear Reviewer wgk8,
> > >
> > > Thank you for your response. We will gladly incorporate all your feedback into the manuscript. We would be grateful to know what points remain, as we aim to fully resolve any outstanding issues and strengthen our work.
> > >
> > > We look forward to your final assessment. Thank you again for your constructive guidance.
> > >
> > > Sincerely,
> > > Authors

---

### Note · Authors · 2025-08-15

We thank the AC and reviewers for their thorough and constructive feedback. Our work, **FunDPS**, addresses the challenge of solving PDE-based inverse problems from sparse measurements by introducing the first principled function-space diffusion framework with theoretical guarantees and strong empirical performance.

Our core contributions are threefold:
- **A Novel Discretization-Agnostic Architecture** using neural operators, which unlocks powerful multi-resolution strategies for robust generalization across grids, flexible plug-and-play guidance, and a **25x inference speedup** compared to existing diffusion-based methods.
- **Theoretical Advancement** via the first generalization of Tweedie’s formula to Banach spaces, enabling rigorous basis for guided sampling in infinite dimensions.
- **State-of-the-Art Performance** achieving a 32% average accuracy improvement over strong baselines on five challenging PDE problems, backed by extensive ablations demonstrating its robustness.

We are very encouraged that the review process has converged to a strong positive consensus. **Four reviewers (8pVy, uwzC, mb5h, dkME) now support acceptance**, with some score raising. The discussions led us to add new experiments on real-world data, further ablation (confirming robustness to guidance weights and extreme data sparsity), and key clarifications, all of which further strengthened the paper.

Regarding the concern on ODE stiffness raised by Reviewer dkME, we respectfully highlight that our generative sampling framework operates under different principles. The extensive empirical evidence across all tasks, supported by clear reviewer consensus, confirms our method's stability and effectiveness, demonstrating its real-world merit.

We believe FunDPS advances PDE-based inverse problem solving by unifying rigorous infinite-dimensional theory, an agile sampling pipeline, and accuracy and speed boost, which provides a robust and efficient design for the SciML community.

We want to thank you again for the valuable insights, many of which will be incorporated into the final manuscript.

---

### Decision · Program_Chairs · 2025-09-17

**Decision:**

Accept (poster)

**Comment:**

The paper introduces a new framework for solving inverse problems for partial differential equations. The main contribution is an extension of a generative diffusion framework—originally developed for jointly modeling the distribution of PDE solutions and their parameter space in finite dimensions—to infinite-dimensional function spaces. The method first trains an unconditional diffusion model in function space and then conditions at inference time on observations. This conditioning leverages a plug-and-play mechanism based on an extension of Tweedie’s formula to infinite-dimensional spaces. The framework implements a neural operator, is resolution-agnostic, and exploits this property to accelerate training. As in the original finite-dimensional diffusion work, the focus is on solving PDE inverse problems under partial observation. Experiments and comparisons are conducted on simulations from a variety of PDEs, addressing both inverse and forward problems.

The reviewers find that the paper makes several contributions, including: a novel function-space formulation of joint distribution modeling, an extension of Tweedie’s formula from finite-dimensional diffusion models to infinite-dimensional spaces, and promising experimental results compared with several baselines across diverse PDE simulations. Some reviewers note, however, that while novel, the originality of the contribution is somewhat limited.

During the rebuttal, the authors provided detailed answers and clarifications, along with new experimental results and ablations. All reviewers agree that most of their concerns were addressed: one increased their score, while the others maintained their initial evaluation.
Considering the discussion, I recommend acceptance. However, in its present form, the paper will be accessible to only a narrow audience. I encourage the authors to provide a clearer and more accessible description of their contribution.